# Functional dissection of inherited non-coding variation influencing multiple myeloma risk

Ram Ajore[1], Abhishek Niroula [1,2], Maroulio Pertesi [1], Caterina Cafaro[1], Malte Thodberg [1], Molly Went[3], Erik L. Bao [2,4,5], Laura Duran-Lozano[1], Aitzkoa Lopez de Lapuente Portilla[1], Thorunn Olafsdottir [6], Nerea Ugidos-Damboriena[1], Olafur Magnusson[6], Mehmet Samur [5], Caleb A. Lareau [2,4,5], Gisli H. Halldorsson [6], Gudmar Thorleifsson[6], Gudmundur L. Norddahl[6], Kristbjorg Gunnarsdottir[6], Asta Försti [7,8], Hartmut Goldschmidt [9], Kari Hemminki[7,10], Frits van Rhee[8], Scott Kimber[3], Adam S. Sperling [5], Martin Kaiser [3], Kenneth Anderson [5], Ingileif Jonsdottir [6], Nikhil Munshi [5], Thorunn Rafnar [6], Anders Waage[11], Niels Weinhold[7,9], Unnur Thorsteinsdottir[6], Vijay G. Sankaran [2,4,5,12], Kari Stefansson [6], Richard Houlston [3] & Björn Nilsson [1,2✉]

Thousands of non-coding variants have been associated with increased risk of human diseases, yet the causal variants and their mechanisms-of-action remain obscure. In an integrative study combining massively parallel reporter assays (MPRA), expression analyses (eQTL, meQTL, PCHiC) and chromatin accessibility analyses in primary cells (caQTL), we investigate 1,039 variants associated with multiple myeloma (MM). We demonstrate that MM susceptibility is mediated by gene-regulatory changes in plasma cells and B-cells, and identify putative causal variants at six risk loci (*SMARCD3*, *WAC*, *ELL2*, *CDCA7L*, *CEP120*, and *PREX1*). Notably, three of these variants co-localize with significant plasma cell caQTLs, signaling the presence of causal activity at these precise genomic positions in an endogenous chromosomal context in vivo. Our results provide a systematic functional dissection of risk loci for a hematologic malignancy.

[1] Hematology and Transfusion Medicine, Department of Laboratory Medicine, BMC B13, 221 84 Lund, Sweden. [2] Broad Institute of Massachusetts Institute of Technology and Harvard University, 415 Main Street, Boston, MA 02142, USA. [3] Division of Genetics and Epidemiology, The Institute of Cancer Research, 123 Old Brompton Road, London SW7 3RP, United Kingdom. [4] Division of Hematology/Oncology, Boston Children's Hospital, Harvard Medical School, Boston, MA, USA. [5] Dana-Farber Cancer Institute, Harvard Medical School, Boston, MA, USA. [6] deCODE Genetics/Amgen Inc., Sturlugata 8, 101 Reykjavik, Iceland. [7] German Cancer Research Center (DKFZ), Im Neuenheimer Feld 580, D-69120 Heidelberg, Germany. [8] Hopp Children's Cancer Center, Heidelberg, Germany. [9] Department of Internal Medicine V, University Hospital of Heidelberg, 69120 Heidelberg, Germany. [10] Faculty of Medicine and Biomedical Center in Pilsen, Charles University in Prague, Prague 30605, Czech Republic. [11] Department of Cancer Research and Molecular Medicine, Norwegian University of Science and Technology, Box 8905, N-7491 Trondheim, Norway. [12] Harvard Stem Cell Institute, Cambridge, MA, USA. ✉email: bjorn.nilsson@med.lu.se

Genome-wide association studies (GWAS) have identified tens of thousands of sequence variants associated with human diseases and traits[1], yet our understanding of the underlying mechanisms is still limited. Each association signal is usually represented by tens to hundreds of variants in linkage disequilibrium (LD). The vast majority of these variants map to noncoding regions of the genome, and likely act by altering gene expression[2–4]. For most signals, however, the causal variants, their target genes, and target cell types remain unknown.

Multiple myeloma (MM) is defined by uncontrolled, clonal growth of plasma cells, usually in the bone marrow. It is a common blood malignancy, with strong epidemiological support for inherited susceptibility[5]. While genome-wide association studies have identified 24 risk loci[6–11], the causal variants remain largely unknown[12,13]. Further, plasma cells can be readily isolated from MM patients using routine methods, and cell lines appropriate for the investigation of MM biology exist. For these reasons, MM is an attractive model disease for deciphering the functional basis of risk loci. To our knowledge, no systematic functional dissection of risk loci for a hematologic malignancy has been reported[14–18].

Here, we carried out an integrative study combining massively parallel reporter assays (MPRA), expression analyses (eQTL, meQTL, and PCHiC), and chromatin accessibility quantitative locus (caQTL) analyses in primary cells to investigate 1039 variants in linkage disequilibrium with multiple myeloma (MM) lead variants. We demonstrate that MM susceptibility is mediated by gene-regulatory changes in plasma cells and B-cells, and identify putative causal variants at six risk loci. Notably, three of these variants co-localize with significant plasma cell caQTLs, signaling the presence of causal activity at these precise positions in an endogenous chromosomal context in vivo. Our results provide a systematic functional dissection of risk loci for a hematologic malignancy.

## Results

**Designing an MPRA to screen MM risk variants.** To identify putative causal variants, we first designed an MPRA[14,19–21] to screen 1039 variants in high LD ($r^2 > 0.8$) with MM lead variants for transcriptional activity (Fig. 1a and Supplementary Table 1). For each variant, we designed twelve 120-bp oligonucleotide sequences corresponding to reference and alternative alleles in six genomic contexts (both strands × three sliding windows with the variant at −20, 0, and +20 bp from the center). Sequences were coupled to a reporter gene with random 20-bp sequence barcodes 3′ of its open reading frame. Following transfection into cell lines, the transcriptional activity of each construct was measured by determining the barcode representation in reporter mRNA relative to DNA (Fig. 1b). Plasmid sequencing identified $1.73 \times 10^6$ unique barcodes tagging 12,378 (99.2%) of the 12,468 designed oligonucleotides (Fig. 1c). As a positive control, we included the *RUNX3* variant rs188468174, which influences immunoglobulin (Ig) levels and exhibits luciferase activity across a broad range of MM cell lines[22].

**Identification of causal cell types for MM susceptibility.** Since reporter assays can show cell type-dependent activity, MPRA should ideally be performed in an appropriate cellular model. We therefore carried out computational analyses to identify cell types where MM risk variants likely act. First, using ATAC-seq data for blood cell populations[23], we found an enrichment of risk variants in genomic regions with accessible chromatin in plasma cells and total mature B-cells (Supplementary Fig. 1). Second, investigating blood cell populations for expression[23–28] of genes located at MM risk loci, we identified plasma cells and total mature B-cells as the

most enriched cell types (Supplementary Fig. 2). Third, using gene expression profiles of CD138+ plasma cells isolated from the bone marrow of 2650 MM patients[12,29–31], we identified *cis*-eQTLs in LD with ten risk alleles (Supplementary Table 2). Additional *cis*-eQTLs were found in whole blood (Supplementary Table 3) or in CD19+ total mature B-cells isolated from 758 random blood donors (Supplementary Table 4). Fourth, since plasma cells are responsible for producing Ig, we tested MM lead variants for association with blood IgA, IgG, and IgM levels[22]. This revealed enrichments of association signal within the set of 24 MM lead variants for all three Ig isotypes (binomial test $P = 6.8 \times 10^{-5}$ for IgA, $P = 0.02$ for IgG, $P = 0.004$ for IgM for the enrichment of association $P$ values <0.05), as well as individually significant associations (Supplementary Fig. 3), including for the *SMARCD3*, *WAC*, and *ELL2* associations, which also showed plasma cell *cis*-eQTLs (Supplementary Table 2). Collectively, these data are consistent with many MM risk variants acting by altering gene regulation in plasma cells, while others may act in other cell populations, including B-cells.

**Identification of MM risk variants influencing transcription.** Focusing our analysis on plasma cells, we performed MPRA in the MM plasma cell lines L363 and MOLP8. Each cell line was assayed in three replicates (Fig. 1d). Based on barcode activity estimates, we calculated a $\log_2$ score for each variant reflecting the transcriptional activity of the alternative relative to the reference allele, averaged across genomic contexts and replicates[32]. L363 and MOLP8 scores showed a positive correlation (Fig. 2a), did not display strand bias (Fig. 2b), and additional validation of 20 selected variants showed a positive correlation with luciferase data (Fig. 2c, Supplementary Fig. 4 and Supplementary Table 5). Moreover, variants with strong MPRA scores were enriched in chromatin accessibility regions in primary plasma cells, consistent with our assay selecting variants with endogenous regulatory activity (Fig. 2d and Supplementary Fig. 5).

In L363, 142 variants were significant (FDR <5%), including 33 with strong effects (absolute $\log_2$ score >0.2). In MOLP8, 28 were significant, including 21 with strong effects (Fig. 2e, f and Supplementary Data 1). The higher number of significant variants in L363, compared to MOLP8, cells was congruent with a higher transfection efficiency (54% for L363 versus 15% for MOLP8) and higher post-transfection viability (90% for L363 versus 65% for MOLP8). In total, 23 variants were significant in both screens, and eight of these showed concordant plasma cell *cis*-eQTLs, making them putative causal variants that were selected for follow-up (Table 1, Fig. 3, and Supplementary Figs. 4, 6). The other 15 had discordant or no plasma cell *cis*-eQTLs, either because of technical limitations (e.g., *TERC* was not in our eQTL data; the *JARID2* and *RUNX3* variants are rare), or because these alter gene expression in another cell state (e.g., *TNFRSF13B* is primarily expressed in switch-memory B-cells[33] and had a *cis*-eQTL in total mature B-cells; Supplementary Table 4).

**Functional characterization of MPRA-functional variants.** We next investigated potential mechanisms of action for the selected variants. rs78740585 maps to *SMARCD3* (Fig. 4a). In humans, *SMARCD1*, *SMARCD2*, and *SMARCD3* encode alternative, mutually exclusive 60-kD subunits of the SWI/SNF nucleosome remodeling complex[34–36]. Incorporation of either SMARCD subunit variant into the complex influences its activity[37]. In blood, *SMARCD3* is primarily expressed in granulocytes and monocytes whereas basal expression in plasma cells is very low; instead these cells exhibit high expression of *SMARCD1* and *SMARCD2* (Fig. 5a). By contrast, the MM risk allele associates

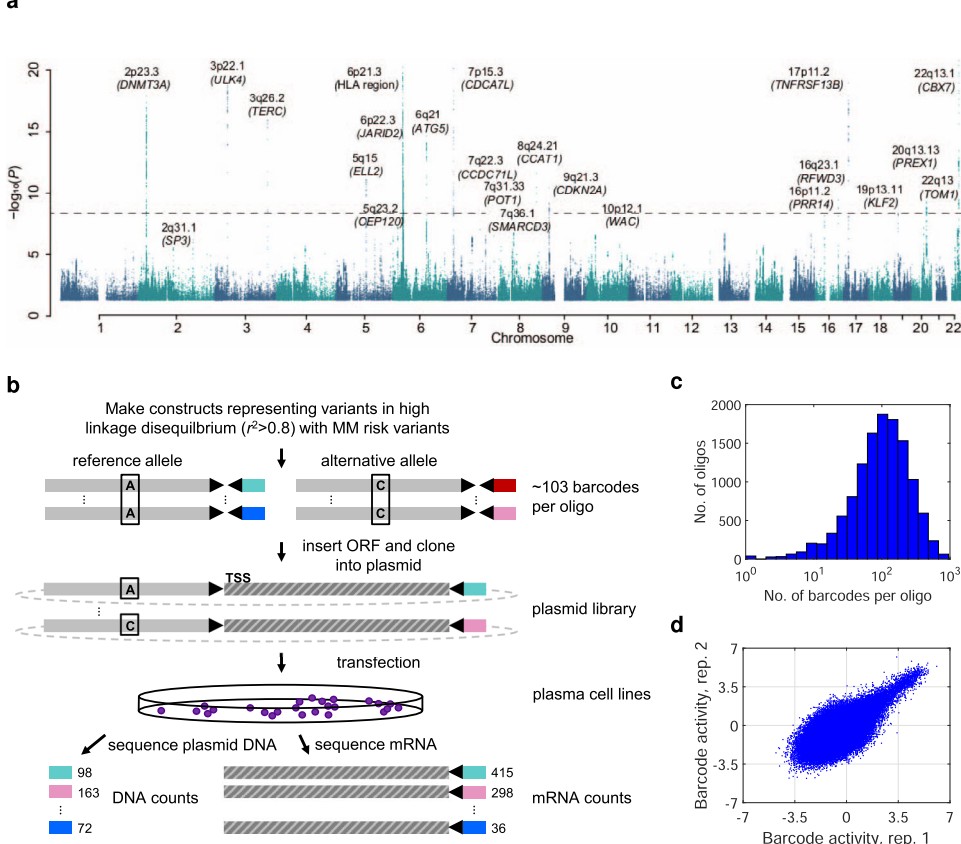

**Fig. 1 Screening assays to identify MM risk variants for transcriptional activity. a** Manhattan plot of the largest genome-wide association study on MM to date, a meta-analysis totaling 9974 MM cases and 247,556 controls of European ancestry[11]. The 23 indicated loci associate with MM at $P < 5 \times 10^{-8}$. The 11q13.3 (*CCND1*) locus specifically associates with risk of t(11;14)[*IGH/CCND1*] translocation MM[56]. Lead variants at each locus are detailed in Supplementary Table 1. **b** We employed MPRA to screen variants in high LD ($r^2 > 0.8$) with MM lead variants for transcriptional activity. For each variant, we designed twelve 120-bp oligonucleotide sequences representing the reference and alternative alleles in six genomic contexts (positive and negative DNA strand × three sliding windows with the variant positioned at −20, 0, or +20 bp from the center). The synthesized oligonucleotides were coupled to a synthetic reporter gene with 20-nt random sequence barcodes 3′ of its open reading frame. **c** Sequencing of the final plasmid library identified $1.73 \times 10^6$ barcodes mapping to 12,378 of the 12,468 designed oligonucleotides. The histogram shows the numbers of barcodes representing each oligonucleotide (median 103). **d** Following transfection of the library into cell lines, the transcriptional activity of each construct was measured by quantifying the barcode representation in reporter mRNA relative to DNA by sequencing. Barcode activity was quantified as $\log_2(1+\#RNA_i)/(1+\#DNA_i)$ where $\#RNA_i$ and $\#DNA_i$ are the read counts for barcode $i$ normalized to counts per 10 million reads. We performed MPRA in three replicates in each cell line. The plot shows the correlation of barcode activity estimates between two L363 replicates.

with upregulation of *SMARCD3* in plasma cells (Supplementary Tables 2 to 4). rs78740585-A creates a binding site for IRF4 (Fig. 5b, c, Supplementary Fig. 7, and Supplementary Data 2), a key plasma cell transcription factor essential for the survival of MM cells[38]. Knockdown of *IRF4* attenuated rs78740585-A luciferase activity (Fig. 5d, e). Furthermore, analysis of promoter-capture Hi-C (PCHi-C) data for three MM plasma cell lines showed a chromatin looping interaction between the rs78740585 region and the *SMARCD3* promoter (Fig. 4a and Supplementary Fig. 8a). Collectively, these data are consistent with rs78740585-A effecting ectopic *SMARCD3* expression in plasma cells by introducing a new IRF4 site into an enhancer, In theory, increased levels of SMARCD3 protein could lead to the displacement of SMARCD1 and SMARCD2 in the SWI/SNF complex through stoichiometric competition, potentially impacting on SWI/SNF-dependent gene expression.

rs2790444 maps to the autophagy gene *WAC* (Fig. 4b). Rare loss-of-function variants in *WAC* cause De Santo-Shinawi syndrome[39], which can feature hypogammaglobulinemia[40]. The common MM risk allele associates with increased levels of IgM in the blood (Supplementary Fig. 3) and downregulation of *WAC* in

plasma cells (Supplementary Table 2). rs2790444 maps close to the *WAC* transcription start site, within the PCHi-C bait region (Fig. 4b and Supplementary Fig. 8b). We found that rs2790444-T creates a binding site for the POU2F1 transcription factor and knockdown of POU2F1 attenuated rs2790444-T luciferase activity (Fig. 6a–d, Supplementary Fig. 9, and Supplementary Data 2). POU2F1 has a dual role in the regulation of gene expression; recruiting the nucleosome remodeling and deacetylase (NuRD) complex, POU2F1 promotes methylation and suppressive histone modifications, while in the context of MAPK signaling it recruits the KDM3A demethylase, promoting pro-transcriptional effects[41]. Consistent with pro-transcriptional activity, we detected a significant plasma cell *cis*-meQTL at *WAC* with rs2790444 ($P = 1.37 \times 10^{-8}$), with rs2790444-T being associated with reduced methylation (Supplementary Table 6). Moreover, CRISPR/Cas9 deletion of a 139-bp region harboring rs2790444 downregulated *WAC*, supporting functional coupling between the variant-harboring region and the transcriptional regulation of *WAC* (Fig. 6e, f). These data are compatible with rs2790444-T creating a promoter-proximal POU2F1 site, upregulating *WAC* through decreased methylation.

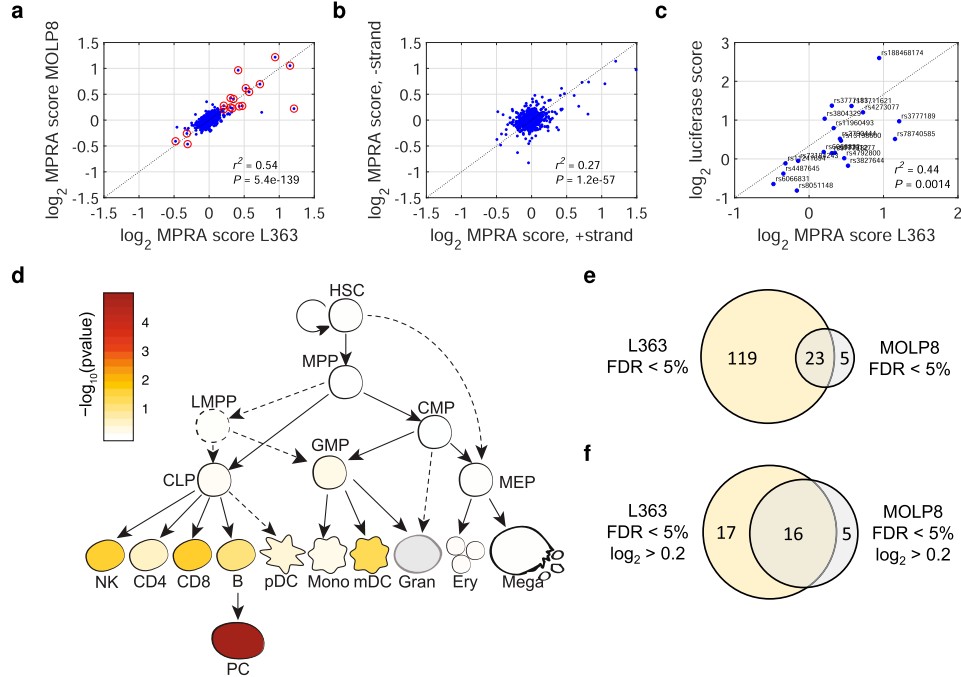

**Fig. 2 Overarching analysis of screening data.** We performed MPRA in the MM plasma cell lines L363 and MOLP8. **a** Variant $\log_2$ scores for the L363 and MOLP8 screens. For each variant, $\log_2$ reflects the transcriptional activity of the alternative relative to the reference allele. Scores were calculated based on barcode activity estimates in all six genomic contexts (i.e., across both strands and all three sliding windows) and three replicates per cell line. Variants with strong effects (absolute $\log_2$ score >0.2) in either screen are indicated in red. Pearson $r$ and two-sided $P$ values are shown. **b** Calculating $\log_2$ scores using either positive-strand (x-axis) or negative-strand constructs (y-axis) for the same variant, we did not observe strand bias. **c** As an additional assay validation step, we carried out luciferase experiments for 20 variants, showing a significant positive correlation between the MPRA effect (x-axis) and the luciferase effect (y-axis). **d** g-chromVAR analysis of screened variants, weighted by their L363 $\log_2$ scores showed enrichment of variants with strong MPRA scores in genomic regions with accessible chromatin in plasma cells, consistent with our assay selecting variants with endogenous regulatory activity. **e** Numbers of significant variants with FDR <5% in the two cell lines. **f** Numbers of variants showing both FDR <5% and strong effects (absolute $\log_2$ score >0.2).

rs3777182, rs3777183, and rs3777189 map to *ELL2* encoding a key protein in the super-elongation complex that drives Ig synthesis[42–45] (Fig. 4c). The MM risk allele downregulates *ELL2* in plasma cells and Ig levels in blood[8,12]. Recently, we nominated rs3777189 as causal using non-systematic approaches and demonstrated that it changes a MAFF/G/K binding site[12]. In our MPRA screen, we now identify rs3777189 as the most active variant within its LD block, providing additional, unbiased evidence for causality. In addition, we identify rs3777182 and rs3777183 as previously unappreciated regulatory variants within the *ELL2* LD block. Analyzing our PCHi-C data, we identified a chromatin looping interaction between the rs3777183-rs3777182 region and the *ELL2* promoter (Fig. 4c and Supplementary Fig. 8c), and predicted several altered motifs (Supplementary Data 2). CRISPR/Cas9 deletion of a 141-bp region harboring rs3777183-rs3777182 and an 89-bp region harboring rs3777189 both altered *ELL2* expression, supporting that the *ELL2* eQTL is caused by genetic variation in multiple intronic regulatory elements that are involved in the transcriptional regulation of *ELL2* (Fig. 7a, b and Supplementary Fig. 10a, b).

rs4487645 maps to the *DNAH11-CDCA7L* locus (Fig. 4d and Supplementary Fig. 8d), and the risk allele rs4487645-C upregulates the cMyc-interacting *CDCA7L*[29,46]. We previously proposed rs4487645 as causal, finding that rs4487645-C creates a new IRF4 binding site[13]. Our current analysis provides additional unbiased evidence for this variant indeed being the functional basis of the 7p15.3 association. CRISPR/Cas9 deletion of a 76-bp region harboring rs4487645 downregulated *CDCA7L* (Fig. 7c and Supplementary Fig. 10c), supporting a regulatory link between the region and *CDCA7L*. Moreover, we employed CRISPR/Cas9 with

homology-directed repair (HDR) to generate L363 single-cell clones with different rs4487645 genotypes. In total, we generated six rs4487645-C-homozygous clones, three rs4487645-C/A heterozygous clones, and six rs4487645-A-homozygous clones. We observed a significant association between rs4487645 genotype and *CDCA7L* expression, with the C allele yielding higher expression (Fig. 7d), further supporting that rs4487645 causes the *CDCA7L* eQTL.

Finally, rs11960493 and rs6066832 map to *CEP120* and *PREX1*, respectively, and upregulate these genes in plasma cells (Fig. 4e, f and Supplementary Fig. 8e, f). *CEP120* is implicated in microtubule assembly[47], and *PREX1* encodes a guanine nucleotide exchange factor mutated or aberrantly expressed in several cancers[48,49]. While we predicted several motif changes for both variants (Supplementary Data 2) and a looping interaction between the rs6066832 region and the *PREX1* promoter (Fig. 4e and Supplementary Fig. 8f), we could not identify differentially bound proteins.

**Effects in the endogenous chromosomal context in vivo.** Following characterization in vitro, we investigated if the eight selected variants are active in an endogenous chromosomal context in vivo. Altered gene-regulatory activity is associated with the release or recruitment of proteins to DNA and/or changes in chromatin structure. In turn, this could cause allele-dependent changes in accessibility (chromatin accessibility quantitative trait loci, caQTLs) around the variant, detectable by ATAC-seq of limited numbers of primary cells. Hence, we performed ATAC-seq on plasma cells from MM patients. To detect caQTLs, we

**Table 1 Variants showing FDR <5% in both L363 and MOLP8 cells, with MM risk alleles underlined.**

| Locus | rsID | Ref | Alt | Gene | L363 cells | | | MOLP8 cells | | | plasma cell cis-eQTL |
|---|---|---|---|---|---|---|---|---|---|---|---|
| | | | | | log$_2$ score | P value | Q value | log$_2$ score | P value | Q value | |
| 5q15 | rs3777189 | C | G | ELL2 | 1.210 | 1.2E-08 | 0.000 | 0.222 | 1.3E-03 | 0.041 | concordant |
| 7q36.1 | rs78740585 | G | A | SMARCD3 | 1.154 | 1.2E-08 | 0.000 | 1.053 | 1.2E-08 | 0.000 | concordant |
| 10p12.1 | rs188468174 | C | T | RUNX3 | 0.941 | 1.2E-08 | 0.000 | 1.217 | 1.2E-08 | 0.000 | n/a[a] |
| 17p11.2 | rs4273077 | A | G | TNFRSF13B | 0.725 | 1.2E-08 | 0.000 | 0.694 | 1.2E-08 | 0.000 | n/a[b] |
| 3q26.2 | rs11711621 | C | T | TERC | 0.570 | 1.2E-08 | 0.000 | 0.544 | 1.2E-08 | 0.000 | n/a |
| 6q21 | rs3827644 | G | C | ATG5 | 0.523 | 1.2E-08 | 0.000 | 0.617 | 1.2E-08 | 0.000 | n/a |
| 20q13.13 | rs6066831 | C | A | PREX1 | −0.479 | 1.2E-08 | 0.000 | −0.409 | 1.2E-08 | 0.000 | discordant |
| 17p11.2 | rs4792800 | A | G | TNFRSF13B | 0.472 | 1.2E-08 | 0.000 | 0.274 | 9.8E-08 | 0.000 | n/a[b] |
| 6p22.3 | rs13198600 | A | G | JARID2 | 0.428 | 1.2E-08 | 0.000 | 0.265 | 1.5E-05 | 0.001 | n/a |
| 10p12.1 | rs27790444 | C | T | WAC | 0.414 | 2.4E-08 | 0.000 | 0.963 | 1.2E-08 | 0.000 | concordant |
| 6q21 | rs77791277 | G | A | ATG5 | 0.347 | 2.4E-08 | 0.000 | 0.408 | 1.2E-08 | 0.000 | n/a |
| 7p15.3 | rs4487645 | C | A | CDCA7L | −0.345 | 1.2E-08 | 0.000 | −0.159 | 1.5E-04 | 0.007 | concordant |
| 5q23.2 | rs11960493 | T | G | CEP120 | 0.330 | 2.4E-08 | 0.000 | 0.227 | 3.0E-04 | 0.012 | concordant |
| 5q23.2 | rs11241694 | T | C | CEP120 | −0.317 | 1.2E-08 | 0.000 | −0.258 | 4.9E-07 | 0.000 | discordant |
| 5q15 | rs3777183 | G | A | ELL2 | 0.306 | 2.4E-08 | 0.000 | 0.426 | 1.2E-08 | 0.000 | concordant |
| 5q15 | rs3777182 | T | A | ELL2 | 0.305 | 2.4E-08 | 0.000 | 0.243 | 2.5E-04 | 0.010 | concordant |
| 5q23.2 | rs6881175 | T | G | CEP120 | −0.296 | 1.2E-08 | 0.000 | −0.159 | 8.0E-04 | 0.026 | discordant |
| 6q21 | rs3804329 | A | G | ATG5 | 0.209 | 2.9E-06 | 0.000 | 0.279 | 2.5E-05 | 0.001 | n/a |
| 20q13.13 | rs6066832 | C | A | PREX1 | 0.197 | 2.4E-08 | 0.000 | 0.210 | 1.9E-06 | 0.000 | concordant |
| 17p11.2 | rs4792798 | G | A | TNFRSF13B | −0.186 | 1.9E-06 | 0.000 | −0.198 | 6.6E-04 | 0.022 | n/a[b] |
| 5q23.2 | rs890913 | A | T | CEP120 | −0.184 | 3.0E-05 | 0.000 | −0.275 | 6.7E-06 | 0.000 | discordant |
| 6p22.3 | rs139637361 | A | G | JARID2 | 0.121 | 2.9E-05 | 0.000 | 0.224 | 4.0E-07 | 0.000 | n/a |
| 5q23.2 | rs6884179 | A | G | CEP120 | −0.105 | 1.4E-03 | 0.012 | −0.177 | 5.8E-04 | 0.021 | discordant |

P- and Q-values are from MPRA score when integrating all barcodes that represent the variant (i.e., all three genomic windows and both strands). Concordant denotes a cis-eQTL in the same direction as the MPRA effect. Discordant denotes a cis-eQTL in the opposite direction.

n/a indicates that no cis-eQTL was detected.

[a]The RUNX3 variant rs188468174, which associates with blood Ig levels, was included as a positive control because of its known luciferase activity in plasma cell lines.

[b]The TNFRSF13B risk allele showed a concordant cis-eQTL in total mature B-cells.

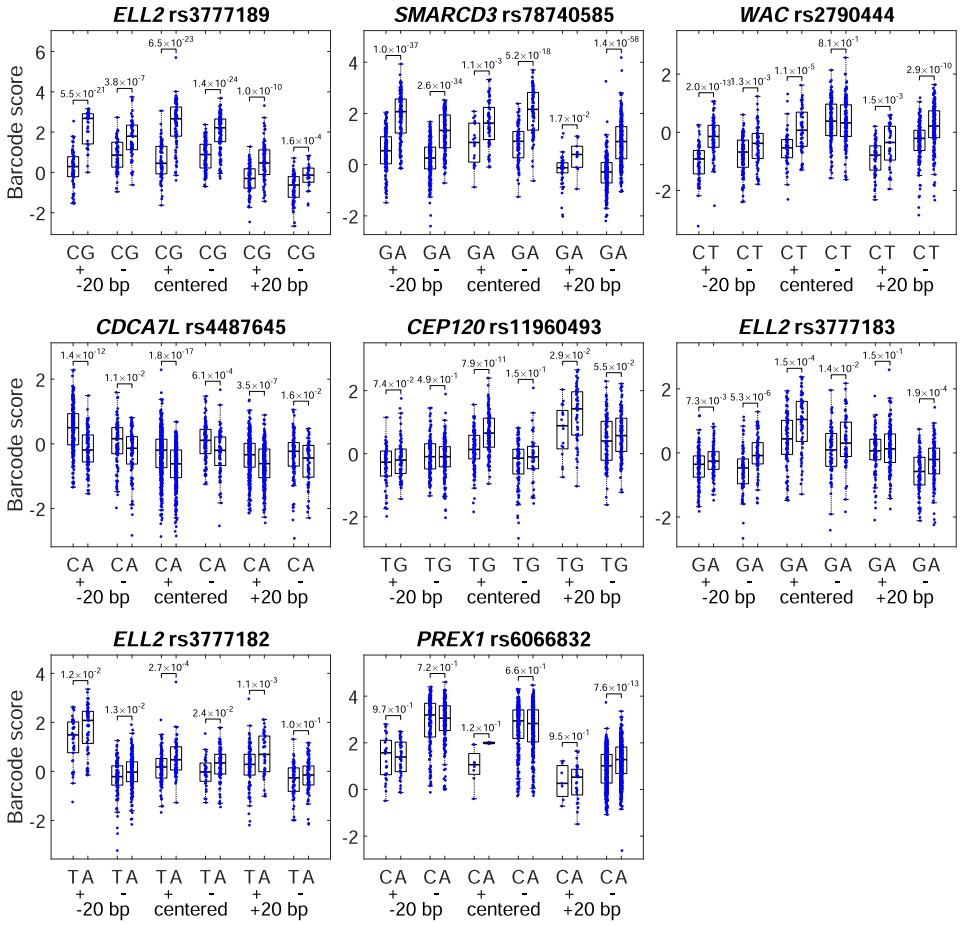

**Fig. 3 MPRA data for identified variants.** Individual MPRA barcode activity estimates for the eight variants in Table 1 also showed concordant plasma cell *cis*-eQTLs. The data have been grouped by allele (reference allele to the left; an alternative to the right), DNA strand (+ or −), and sliding window (variant at −20, 0, or +20 bp from the center of the 120-bp oligonucleotide representing the genomic context). Blue dots represent individual barcode activity estimates. The bottom, middle, and top of each box plot represent the 25th, 50th, and 75th percentiles. The whiskers represent the non-outlier minimum and maximum values, located at 1.5 times the interquartile range from the bottom and top of the box, respectively. The numbers by the brackets are *P* values for the two-sided Student's *t*-test. The luciferase validation data for these eight variants are shown in Supplementary Fig. 4. The individual barcode activity estimates for MOLP8 cells, as well as individual barcode activity estimates for the other variants in Table 1, are shown in Supplementary Fig. 6.

estimated the local ATAC-seq signal intensity as the average Tn5 transposase cut-site density across a 150-bp sliding window positioned at every 10 bps across LD regions and examined correlations with the MM lead variant. We also developed a segmentation algorithm ("caQTLseg") to partition LD regions into subregions with either allele-dependent or allele-independent accessibility.

In an initial set of 56 ATAC-seq samples, we detected lead variant caQTLs at *SMARCD3*, *CDCA7L*, and *CEP120* (Supplementary Fig. 11). For replication, we performed ATAC-seq on an additional 105 samples. In a combined analysis of all 161 samples, the three caQTL signals increased in significance (Fig. 8). The regions identified as having allele-dependent accessibility were identified with a broad range of caQTLseg parameter settings (Online Methods and Supplementary Figs. 12 and 13). Furthermore, the *SMARCD3* and *CDCA7L* signals were centered at rs78740585 and rs4487645, and were the only LD variants within their caQTLs; both of these risk variants create new IRF4 binding sites. Consistent with the recently described role of IRF4 as a pioneer-like transcription factor that regulates chromatin accessibility[50–53], the *SMARCD3*- and the *CDCA7L*-high-expressing MM risk alleles associated with increased accessibility at rs78740585 and rs4487645 (Fig. 8a, b). By contrast, the caQTL at

*CEP120* (Fig. 8c) was more complex, encompassing rs11960493 plus eight other LD variants, one of which (rs62376437) was borderline-significant in the MPRA (*q* value $7.57 \times 10^{-6}$ in L363; $2.72 \times 10^{-1}$ in MOLP8; Supplementary Data 1) and concordant with the *CEP120 cis*-eQTL, suggesting multi-variant causality, as in the case of the *ELL2* association. These results demonstrate that three of our selected MPRA-functional variants co-localize with significant plasma cell caQTLs, signaling the presence of causal regulatory activity at these variants in an endogenous chromosomal context in vivo.

## Discussion
We have carried out a systematic functional analysis of inherited noncoding variants that predispose for MM. Our analysis represents a functional dissection of inherited noncoding variation predisposing for a hematologic malignancy. To our best knowledge, MPRA and caQTL analysis have not been previously used as mutually complementary approaches to identify putative causal variants. While MPRA is a powerful in vitro screening approach, caQTLs provide evidence for causal regulatory activity at specific genomic positions in an endogenous chromosomal context in vivo.

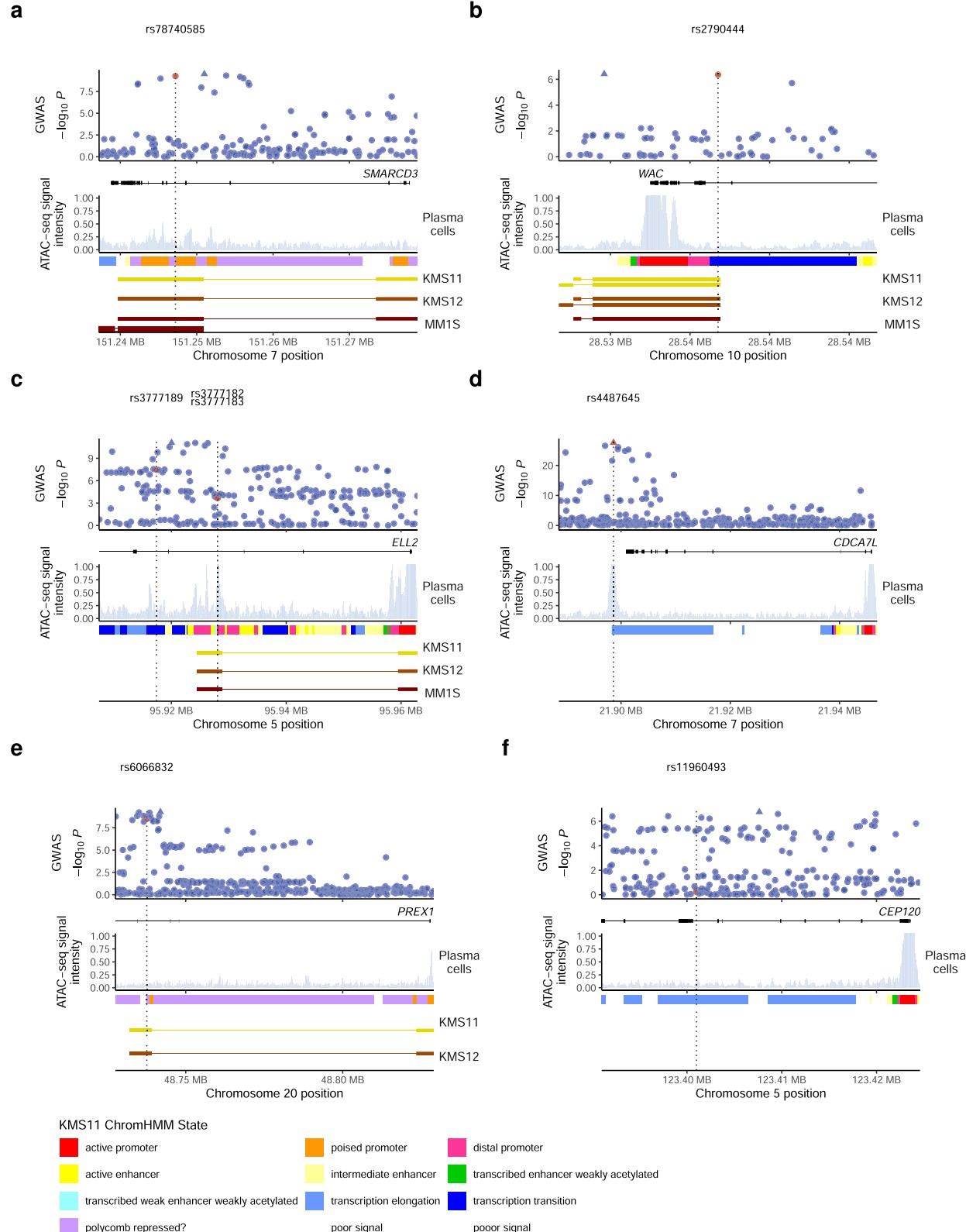

Our analysis identifies eight putative causal regulatory variants at six risk loci: *SMARCD3*, *WAC*, *ELL2*, *CDCA7L*, *CEP120*, and *PREX1*. Out of these variants, seven map to intronic regions within their target genes, and one maps to an enhancer region within a neighboring gene (Fig. 4). These observations are in accordance with other studies where GWAS signals have been dissected functionally (*c.f.*, refs. [14,16,19,23,54,55]). Notable findings include a variant effecting ectopic expression of the SWI/SNF gene *SMARCD3* in plasma cells by introducing a new IRF4 site into an enhancer, and a variant upregulating the autophagy gene *WAC* by creating a POU2F1 site. Additionally, we find evidence for multi-variant causality at *ELL2* and *CEP120*, and further

**Fig. 4 Genomic context of identified putative causal variants.** Based on our functional screens, we identified eight putative causal variants (highlighted in red and with dashed lines) across six loci. The figure shows their association *P* values (Fig. 1a), with the lead SNP indicated as a triangle, along with ATAC-seq data for plasma cells (blue) and 11 ChromHMM states in the MM plasma cell line KMS11. We also generated PCHi-C data for the MM plasma cell lines KMS11 (yellow), KMS12 (orange), and MM1S (red), and identified chromatin looping interactions using the CHiCAGO tool. Interactions with −log$_{10}$(CHiCAGO *P* score) ≥2 are shown. **a** At the *SMARCD3* locus, we detected a chromatin looping interaction between the rs787404585 region and the *SMARCD3* promoter. **b** rs2790444 at *WAC*, located close to the promoter within the PCHi-C bait region. **c** rs3777189 and rs3777183-rs3777182 at *ELL2*, where rs3777182 and rs3777183 are located only 17 bp apart. We detected a chromatin looping interaction between the rs3777183-rs3777182 region and the promoter. **d** No looping interactions were detected at *CDCA7L*. **e** At the *PREX1* locus, we detected a looping interaction between the rs6066832 region and the *PREX1* promoter. **f** No looping interactions were detected for the *CEP120* association. Vertical lines indicate variant positions. Coordinates are hg38.

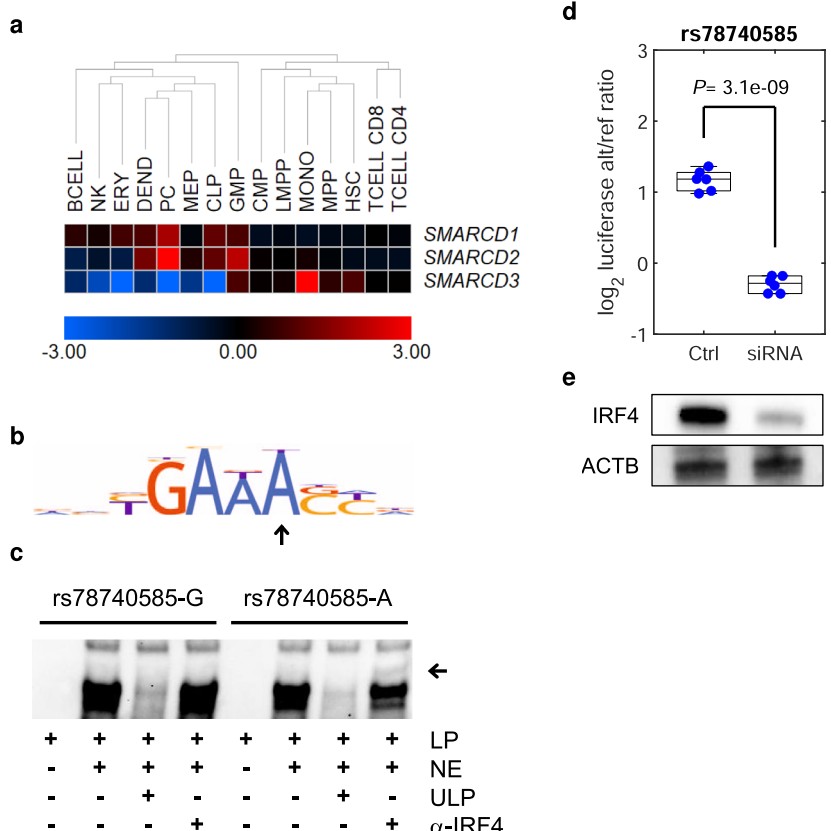

**Fig. 5 Characterization of rs78740585. a** Heat map showing the expression of the *SMARCD* gene family in blood cells. Notably, expression of *SMARCD3* in plasma cells is normally very low; the MM risk allele upregulates *SMARCD3* in this cell type (Supplementary Table 2). The color scale is log$_2$-transformed, median-centered RNA-seq data[23]. **b** Motif analysis predicted that the *SMARCD3* high-expressing rs78740585-A allele creates a binding site for IRF4. Arrow indicates the altered recognition base. **c** Electromobility shift assay showing selective binding of IRF4 to rs78740585-A probe. Arrow indicates supershift with an antibody towards IRF4. **d** siRNA-knockdown of IRF4 reduced luciferase activity for rs78740585-A relative to rs78740585-G in L363 cells. The y-axis indicates the log$_2$ ratio of the luciferase/renilla signal for rs78740585-A relative to the rs78740585-G construct. The bottom, middle, and top of each box plot represent the 25th, 50th, and 75th percentiles. The whiskers represent the non-outlier minimum and maximum values, located at 1.5 times the interquartile range from the bottom and top of the box, respectively. The *P* value is for the two-sided Student's *t*-test. **e** Western blot confirming knockdown. LP labeled probe, NE L363 nuclear extract, ULP unlabeled probe, α-IRF4 antibody against IRF4.

support for rs4487645 being a causal variant at *CDCA7L*. Collectively, our findings provide functional insight into the genetic architecture of MM predisposition.

Regarding limitations, functional dissection of a GWAS signal should ideally include systematic perturbation of each variants within the LD block, for example using CRISPR-HDR or base editors (to replace each reference allele with its corresponding variant allele or vice versa in situ). However, it is widely recognized that such an approach is currently not possible, both because of the workload and because only some variants are accessible to precision editing because of the lack of nearby sgRNAs, and base editors can only achieve certain types of base changes. Additionally, in the case of MM, it

is not possible to culture primary plasma cells or primary multiple myeloma cells ex vivo, and thus any editing experiments will need to be done in cell lines. For these reasons, we instead followed up our MPRA screen with dual-sgRNA CRISPR/Cas9 experiments to link variant-harboring regions to eQTL target genes. We achieved successful editing of rs4487645 at *CDCA7L*, whereas precision editing was not achieved for the other variants of interest. Finally, we carried out caQTL experiments in primary MM plasma cells, demonstrating allele-dependent chromatin accessibility (as a sign of altered regulatory activity) at the positions of the *SMARCD3*, *CDCA7L*, and *CEP120* MPRA-functional variants in an endogenous chromosomal context.

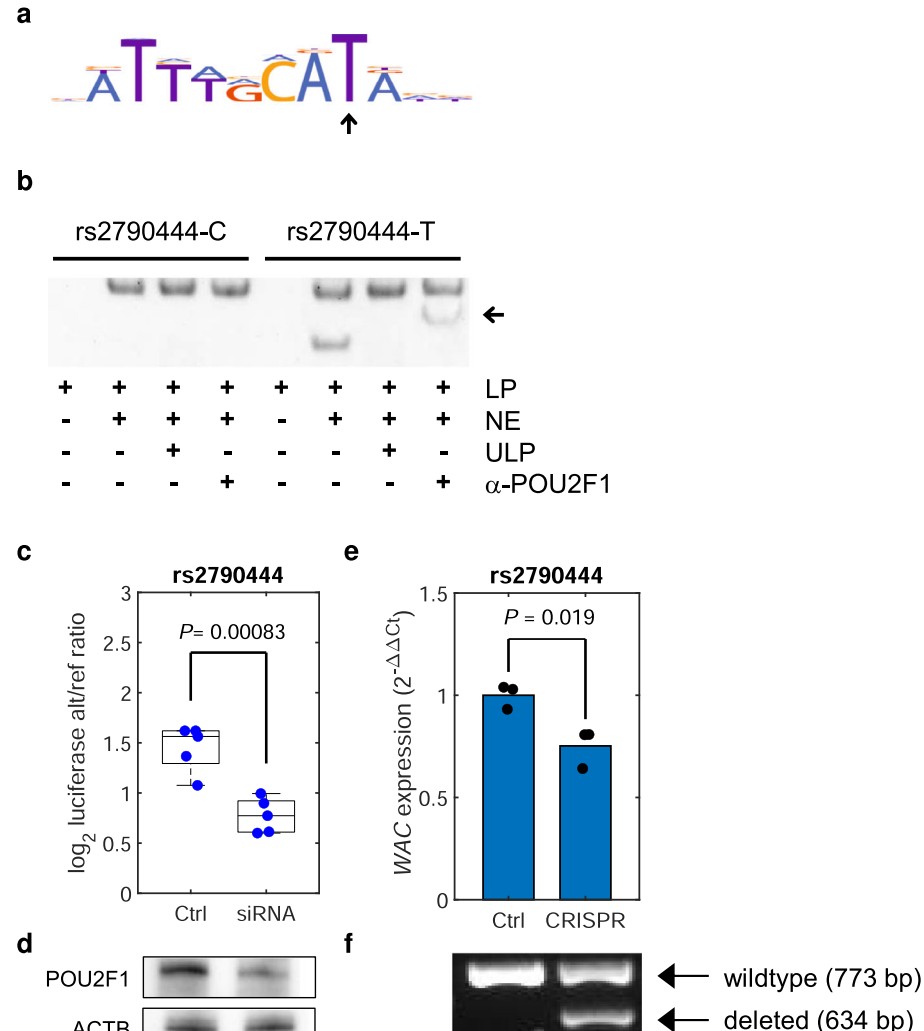

**Fig. 6 Characterization of rs2790444. a** Motif analysis predicted that the *WAC* high-expressing allele rs2790444-T creates a new binding site for POU2F1. Arrow indicates the altered recognition base. **b** Electromobility shift assay showing selective binding of POU2F1 to rs2790444-T probe. Arrow indicates supershift with an antibody towards POU2F1. **c** siRNA-knockdown of POU2F1 attenuated luciferase activity for rs2790444-T relative to rs2790444-C in L363 cells. The *y*-axis indicates the log₂ ratio of the luciferase/renilla signal for rs2790444-T relative to the rs2790444-C construct. The bottom, middle, and top of each box plot represent the 25th, 50th, and 75th percentiles. The whiskers represent the non-outlier minimum and maximum values, located at 1.5 times the interquartile range from the bottom and top of the box, respectively. **d** Western blot confirming knockdown. **e** Quantitative PCR showing expression of *WAC* relative to control in MOLP8 cells upon dual-sgRNA CRISPR/Cas9 deletion of a 139-bp region harboring rs2790444, heterozygous for rs2790444. Blue bars indicate the averages of the individual measurements. **f** Agarose gel confirming deletion of the targeted region. LP labeled probe, NE L363 nuclear extract, ULP unlabeled probe, α-POU2F1 antibody against POU2F1. The *P* values is for the two-sided Student's *t*-test.

Deciphering the functional basis of cancer risk variants provides for a more comprehensive understanding of the biological networks underlying tumorigenesis and predisposition. Here we have addressed this challenge in the context of MM by combining information from high-throughput functional screens, QTL analyses, and additional assays. Our integrative approach illustrates how functional dissection of noncoding variation influencing the development of human malignancies can be undertaken.

## Methods

**MPRA.** We designed an MPRA for variants in LD ($r^2 > 0.8$) with lead variants at 22 loci robustly associated with MM risk (Supplementary Table 1)[5–9,11,56]. We also included twelve candidate MM risk loci[8,11,57] that have not so far been replicated[5,11,58], although these were excluded in the final data analysis. Finally, we included *RUNX3* rs188468174 as a positive control because of its known luciferase activity in plasma cell lines[22].

For each variant, we designed twelve 120-bp sequences corresponding to the reference and alternative allele in six genomic contexts (positive and negative strand × three windows with the variant at −20, 0, and +20 bp from the center),

flanked by 15-bp adapters: [5′-ACTGGCCGCTTGACG-(oligo)-CACTGCGGCTC CTGC-3′]. In total, 12,468 sequences representing 1039 variants were synthesized (CustomArray Inc.). Random 3′ 20-bp barcodes were then added by PCR (Supplementary Table 7). The library was synthesized per ref. [19]. Barcoded oligos were inserted by Gibson assembly (cat no. E2611S, New England Biolabs) into a pGL4:23:ΔxbaΔluc vector to create a mpraΔorf library. A mpra:gfp library was then generated from the mpraΔorf library by inserting minimal promoter, GFP, and partial 3′ UTR from Pgl4.23:gfp plasmid (gift from Ryan Tewhey[19]). The final library was transfected (Neon system; Life Technologies) into $5 \times 10^8$ L363 or MOLP8 cells (ACC49 and ACC569; DSMZ). Cells were cultured at 37 °C and 5% $CO_2$ in RPMI 1640(1X) + GlutaMAX with 10% fetal bovine serum (Gibco BRL, Thermo Fisher Scientific) at 0.5 to $0.7 \times 10^6$ cells/mL. 48 h after transfection, RNA was extracted and reporter mRNA pulled down. After adding sequencing adapters to cDNA synthesized from the DNase-treated GFP-mRNA, samples were sequenced (Illumina NextSeq 1 × 75 bp).

**eQTL and gene expression data**. To identify *cis*-eQTLs in plasma cells, we analyzed gene expression profiles of CD138⁺ cells isolated from bone marrow aspirates from MM patients harvested using immunomagnetic beads. First, we used Affymetrix microarray data, including 183 UK Myeloma IX trial patients (a study aimed at comparing two bisphosphonates in the treatment of MM; Medical

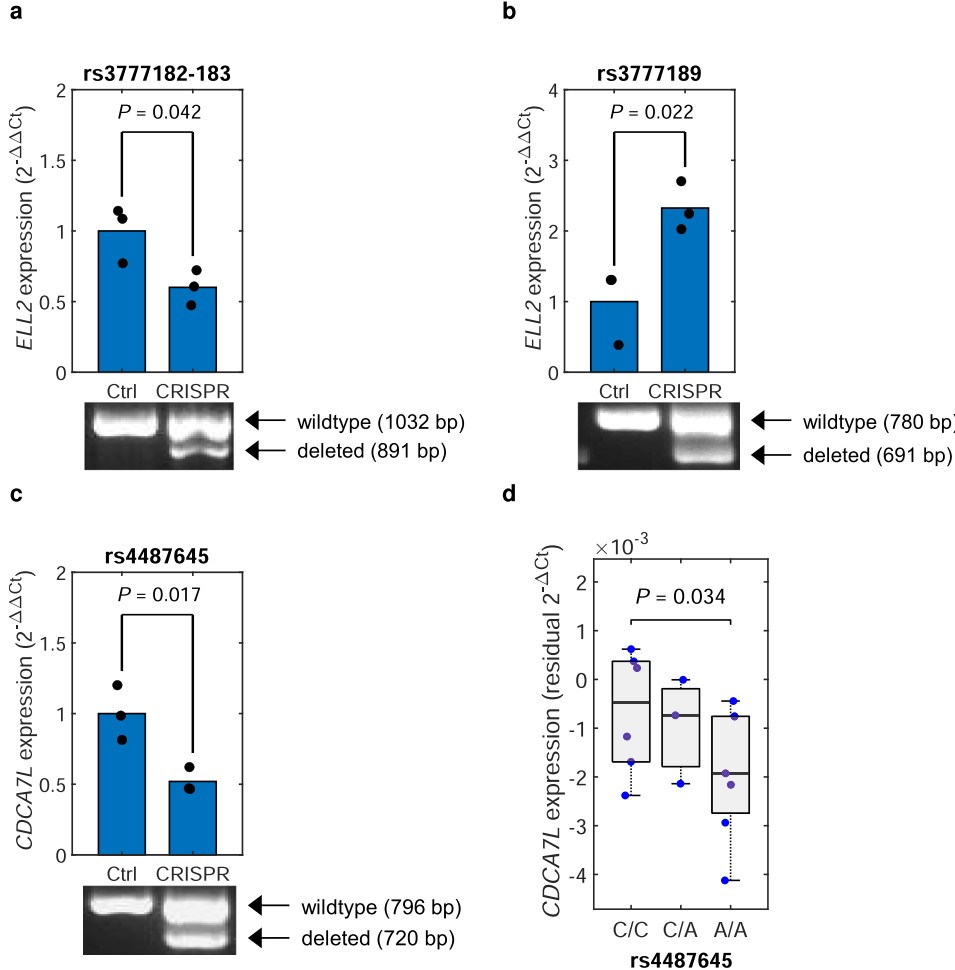

**Fig. 7 Deletion data for rs3777182, rs3777183, and rs3777189 at *ELL2* and rs4487645 at *CDCA7L*. a** Quantitative PCR data showing altered expression relative to control of *ELL2* upon dual-sgRNA CRISPR/Cas9 deletion of a 141-bp region harboring rs3777182 and rs3777183 in RPMI-8226 cells, which are heterozygous the rs3777189, rs3777182, and rs3777183 variants. The agarose gel below confirms the deletion of the CRISPR/Cas9-targeted region. **b** Corresponding data for an 89-bp region harboring rs3777189. **c** Corresponding data for a 76-bp region harboring rs4487645 in OPM2 cells, which are homozygous for the rs4487645-C variant. The *P* values are for the two-sided Student's *t*-test. Blue bars in (**a**) through (**c**) indicate the averages of the individual measurements. **d** We successfully edited rs4487645[C > A] in L363 cells using CRISPR-HDR. We generated six C-homozygous, three C/A heterozygous, and six A-homozygous single-cell clones. Consistent with the other data for rs4487645, we detected an association between CRISPR-edited rs4487645 genotype and *CDCA7L* expression, with the C allele yielding higher expression, further supporting causality. The *y*-axis indicates residual *CDCA7L* expression qPCR expression value in L363 cells, taking into account the *CDCA7L* copy number in each single-cell clone. The bottom, middle, and top of each box plot represent the 25th, 50th, and 75th percentiles. The whiskers represent the non-outlier minimum and maximum values, located at 1.5 times the interquartile range from the bottom and top of the box, respectively. The *P* value is for correlation between edited genotype and *CDCA7L* expression, taking *CDCA7L* DNA copy number into account as a covariate.

Research Council Leukemia Data Monitoring and Ethics committee, no. MREC 02/8/95, ISRCTN68454111)[59], 658 German GMMG patients, and 604 patients treated at the University of Arkansas for Medical Sciences Myeloma Center, USA[11]. Second, we used 185 RNA-seq samples from Lund University (Lund, Sweden)[12]. Third, we used 716 RNA-seq samples with DNA copy number covariates from the CoMMpass study[31]. Fourth, we used 309 RNA-seq samples from the Dana Farber Cancer Institute (Boston, USA)[30]. For the first two data sets, paired SNP microarray genotypes were available. For the third and fourth data sets, only RNA-seq data were available, limiting eQTL analysis to risk alleles with these coding proxies: rs3815768, rs34562254, rs6122720, rs1052501, rs7193541, and rs7782699. For blood, we used eQTLgen (www.eqtlgen.org) and data at deCODE Genetics (RNA-seq for 13,175 Icelanders). For B-cells, we generated eQTL data for 758 Icelanders by isolating B-cells from peripheral blood with negative selection using magnetic beads (StemCell Technologies 19674). To test for enrichment of gene expression of MM-associated genes in blood cell populations, we used gene expression microarray data for sorted blood cells (NCBI Gene Expression Omnibus; accession GSE24759, GSE15695, GSE4581, GSE19784, GSE26760, and GSE5900). These were generated on Affymetrix microarrays and quantile-normalized to a log-normal distribution. For enrichment testing, we used a one-sided Student's *t*-test for genes in MM-associated regions versus other genes in the genome.

**MPRA data analysis**. To map oligo-barcode combinations, we amplified the mpraΔorf library using Illumina_Universal_Adapter and MPRA_v3_TruSeq_Amp2Sa_F primers, added indices by PCR using Illumina_Universal_Adapter and Illumina_Multiplex primers and sequenced the library (Illumina HiSeq 2 × 150 bp). Paired-end reads were merged using PEAR (v0.9.10)[60] and aligned to the designed sequences using BWA-MEM (v0.7.15)[61]. Alignments with more than four mismatches within the designed oligonucleotide, or mismatches within 10 bp of the variant, were excluded. Based on filtered alignments, oligonucleotide-barcode pairs were identified. Combinations supported by at least two reads were included in the mapping. However, barcodes that mapped to more than one oligonucleotide were discarded if fewer than 50 reads supported the barcode or none of the oligo-barcode combinations were supported by at least 95% of the reads (i.e., if one oligo-barcode combination was supported by at least 95% of more than 50 reads, that combination was included). In total, we identified $1.73 \times 10^6$ oligonucleotide-barcode pairs mapping to 12,378 of the 12,468 designed sequences.

To score variants, we used MPRA score[32]. Basically, the activity of each barcode was estimated based on $b_i = \log_2(1 + \#RNA_i)/(1 + \#DNA_i)$, where $\#RNA_i$ and $\#DNA_i$ are the read counts for barcode *i* normalized to counts per 10 million reads. Subsequently, an overall $\log_2$ score representing the transcriptional activity of the alternative relative to the reference allele was calculated by forming the weighted

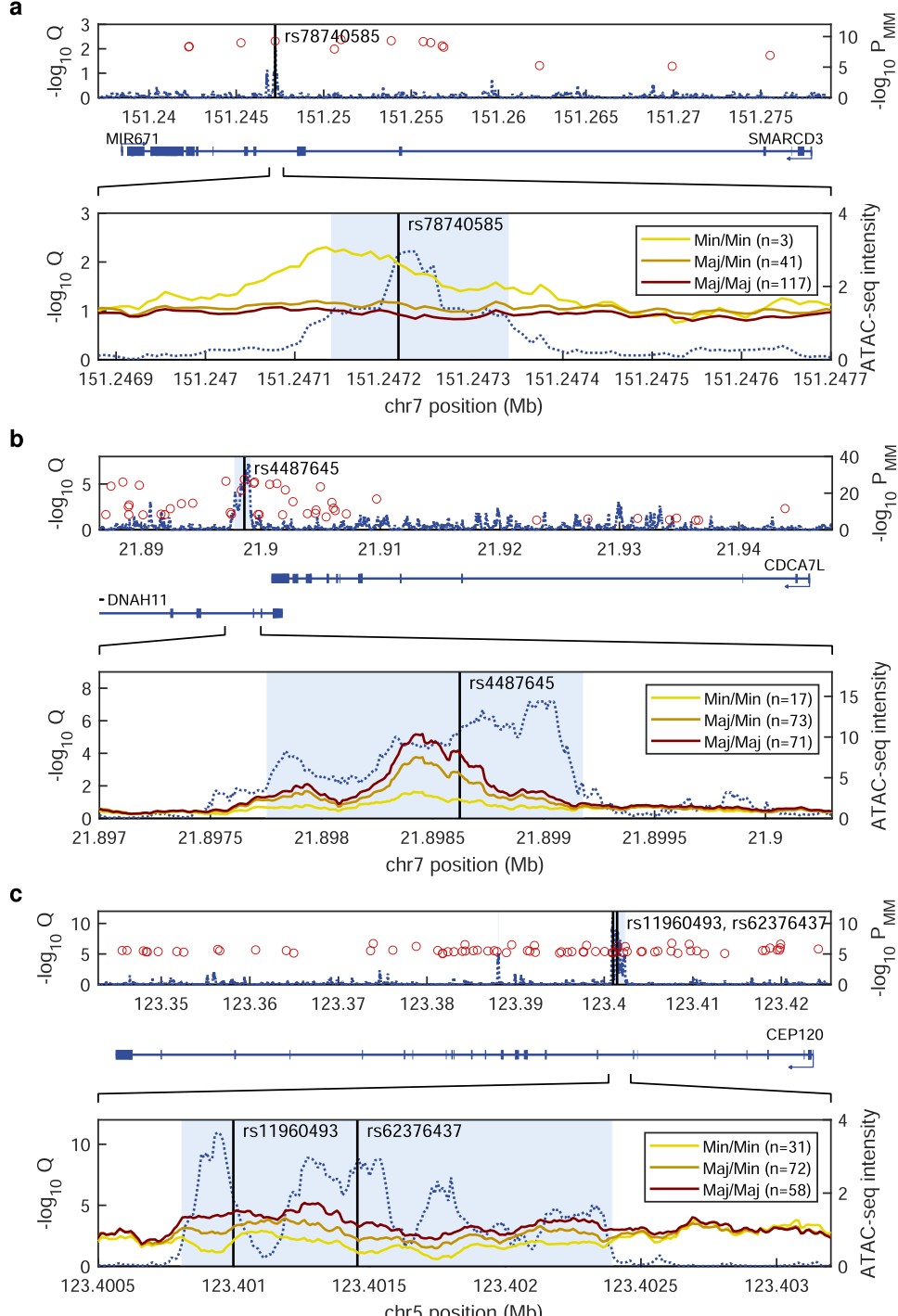

**Fig. 8 Identification of co-localized caQTLs at MPRA-functional variants.** We performed ATAC-seq on plasma cells from 161 MM patients and scanned the LD regions for lead variant caQTLs using two computational approaches. **a** In the *SMARCD3* region, we detected a significant caQTL around rs78740585, with the minor allele conferring increased accessibility. Consistent with this, rs78740585[T > A] showed a positive MPRA log$_2$ score and the rs78740585-A risk allele creates an IRF4 site (Supplementary Fig. 4). **b** In the *CDCA7L* region, we detected a significant, 1.6 kb wide caQTL around rs4487645, with the major allele conferring increased accessibility. Consistent with this, rs4487645[C > A] showed a negative MPRA log$_2$ score and the rs4487645-C risk allele creates an IRF4 site. **c** At *CEP120*, we detected a significant caQTL covering rs11960493 and eight other variants, including rs62376437 which was borderline-significant in MPRA, suggesting the *CEP120* association is enshrined in multiple causal variants. Dashed blue indicates a false discovery rate ($-\log_{10} Q$ value) for Pearson correlation between ATAC-seq signal intensity and lead variant genotype. Regions with lead variant-dependent accessibility called by caQTLseg are indicated in light blue. Upper panels show full regions of LD, lower panels are close-ups of highlighted regions. Red circles indicate variants that show evidence of association with MM (data from Fig. 1a; variants with $P < 10^{-5}$ for association shown). In the lower panels, average local ATAC-seq signal intensity across individuals with different lead variant genotypes is indicated by the yellow (minor/minor), orange (minor/major), and red (major/major) lines.

average of the $b_i$ belonging to the variant across the six genomic contexts and three replicates[32]. To identify strand bias, we also calculated $\log_2$ scores based on constructs representing the variant on either the positive or negative strand.

**ATAC-seq data for blood cell populations**. Sequencing reads for published ATAC-seq libraries from 18 sorted hematopoietic cell types were downloaded from the Sequence Read Achieve[23,62]. Reads were processed as the MM ATAC-seq libraries using the hg38 reference genome. Next, we created a master peak file by aggregating the summits of each population and enumerating the fragments overlapping each peak for each population[62]. From this peak-by-cell type matrix, we performed g-chromVAR[23] to discern cell type enrichments using two types of annotations for the MM variants: the fine-mapped probability of causality (Supplementary Fig. 1) and $\log_2$ MPRA scores (Fig. 2d).

For the first case, we used recalibration of marginal association effects using an approximate Bayes' method[63] as a proxy for fine-mapping to obtain a probability of causality for each MM risk variant[11]. Because the approximate Bayes' method does not account for LD, we first performed stepwise conditional analysis, where we did not detect any secondary signals at any of the loci[8,9,11]. We intentionally used a pure genetics approach, as opposed to fine-mapping methods that factor in functional annotations, to ensure that the downstream g-chromVAR cell type enrichment analysis would be unbiased. For the second case, we used the MPRA score $\log_2$ scores to weight variants by the strength of transcriptional activity. g-chromVAR $p$ values thus correspond to the enrichment of MM risk variants within cell types, weighted by quantitative chromatin accessibility signatures and either the variant genetic fine-mapping score or $\log_2$ MPRA score. Default parameters for g-chromVAR were used.

**meQTL data generation and analysis**. We performed *cis*-meQTL analysis using Illumina 450 K methylation array data for plasma cells from 379 patients from the MRC Myeloma XI trial[64]. Briefly, patients were randomized to induction therapy with CTD (cyclophosphamide, thalidomide, and dexamethasone) or CRD (cyclophosphamide, lenalidomide, and dexamethasone) with or without CVD (cyclophosphamide, bortezomib, and dexamethasone) intensification in patients with less than very good partial response (VGPR) after initial CRD or CTD. Fitter, younger patients were included in the intensive treatment pathway and received high-dose melphalan (HD-MEL) and autologous stem cell transplantation (ASCT) as consolidation. Post induction ± ASCT patients were randomized to lenalidomide, lenalidomide plus vorinostat, or observation. Primary outcome data has been reported. The collection of samples was undertaken with informed consent and ethical review board approval from the Oxfordshire Research Ethics Committee (MREC 17/09/09, ISRCTN49407852). Diagnosis of MM was established in accordance with World Health Organization guidelines. MM cells from patient bone marrow aspirates were obtained at diagnosis and purified (>95%) using immunomagnetic beads with CD138 antibody (Miltenyi Biotec). RNA and DNA were extracted using RNA/DNA mini kit or Allprep kits (Qiagen). The EZ DNA Methylation kit (Zymo Research) was used for bisulfite conversion of genomic DNA. Tumor DNA methylation was profiled using Illumina Infinium Human-Methylation450 (450k) or EPIC 850 K arrays. Raw data were exported from Genome Studio (Illumina) and quality checking and normalization was performed using the ChIP Analysis Methylation Pipeline (ChAMP)[65]. The BMIQ method was used to perform normalization. Preprocessed data were analysed using a Bayesian approach to the probabilistic estimation of expression residuals to infer broad variance components, thus accounting for hidden determinants influencing global expressions such as copy number, translocation status, and batch effects[66]. Genetic associations were tested under an additive model between variants and normalized methylation probes using FastQTL[67], adjusting for plate and methylation-based principal component analysis score[67].

**PCHi-C data generation and analysis**. To identify interactions between variant-harboring regions and promoters, we analyzed published PCHi-C data for KMS11 cells[68,69]. Additionally, we generated PCHi-C data for two additional MM plasma cell lines, KMS12 and MM1S, using the same protocol[68,69].

Briefly, KMS11, KMS12, and MM1S cell lines were obtained from the American Type Culture Collection (ATCC). All cell lines were cultured at 37 °C, in RPMI supplemented with 10% FBS. To generate PCHi-C libraries, 25 million cells were fixed in 1% formaldehyde for 10 min. Cross-linked DNA was digested using HindIII (NEB; #R0104). Digested chromatin ends were filled and marked with biotin-14-dATP (Thermo Fisher, 19524-016). The resulting blunt-ended fragments were ligated at 16 °C in the nucleus with T4 DNA ligase (NEB; #M0202) to minimize random ligation. DNA was de-cross-linked by proteinase K (Ambion; #AM2546) treatment. DNA was sheared by sonication (Covaris; #M220) and 200–650-bp fragments were selected. Biotin-tagged DNA was pulled down with streptavidin beads and ligated with Illumina paired-end adapters (Illumina). Six cycles of PCR were performed to amplify libraries before capture. Promoter-capture was based on 32,313 biotinylated 120-mer RNA baits (Agilent Technologies) targeting both ends of HindIII-restriction fragments that overlap Ensembl promoters of protein-coding, noncoding, antisense, small nuclear RNA, microRNA, and small nucleolar RNA transcripts. A post-capture PCR amplification step was carried out using five amplification cycles, after library

enrichment. Libraries were sequenced using Illumina HiSeq 2000 technology. Reads were aligned to the GRCh37 build using Bowtie2 v.2.2.640 and identification of valid read pairs was performed using HiCUP v.0.5.941. To call significant contacts, HiCUP output was processed using CHiCAGO v.1.1.842. For each cell line, data from three independent biological replicates were combined to obtain a definitive set of contacts. Looping interactions were called using the CHiCAGO pipeline[70] to obtain a unique list of reproducible contacts. Interactions with $-\log_{10}$(CHiCAGO $P$ score) $\geq 2$ were considered significant and shown in figures. Genomic loci or genome browser figures were generated using tidyGenomeBrowser (https://github.com/MalteThodberg/tidyGenomeBrowser). Transcript models were obtained from the TxDb.Hsapiens.UCSC.hg38.knownGene R-package[71]. ChromHMM states for KMS11 cells were obtained from ref. [11] in hg19 coordinates and converted to hg38 coordinates using the rtracklayer R-package[72].

**Luciferase analysis**. Luciferase constructs were generated by cloning genomic sequences (Integrated DNA Technologies; Supplementary Table 5) centered on variants of interest into the pGL3-basic vector. Using electroporation (Neon system; Thermo Fisher Scientific), the constructs were co-transfected with renilla plasmid to enable normalization of the luciferase signal. At 24 h post-electroporation, luciferase and renilla activity was measured using DualGlo Luciferase (cat no. E1960; Promega) on a GLOMAX 20/20 Luminometer. Based on luciferase/renilla readings, we calculated $\log_2$ scores for each variant reflecting the luciferase activity of the alternative relative to the reference allele.

**Transcription factor motif analysis**. To identify differentially binding transcription factors, we used the PERFECTOS-APE tool (http://opera.autosome.ru/perfectosape) with the HOCOMOCO-10, JASPAR, HT-SELEX, SwissRegulon, and HOMER motif databases.

**Electrophoretic mobility shift assays**. For each variant 25-bp 5′-biotin-labeled, double-stranded probes were synthesized (Integrated DNA Technologies): 5′-ACTTAATTTGCC[C/T]GAATTACATTTC-3′ for rs2790444; 5′-TCAA-GAACTGAA[G/A]CTGTAAGTTGAC-3′ for rs78740585. Unlabeled identical sequences were synthesized for competition reactions, and the nuclear extract was prepared[12,73]. For supershift reaction, we used antibodies against POU2F1 (cat no. sc-8024; Santa Cruz) and IRF4 (cat no. 646412; Biolegend). Reaction mixes were incubated for 15 min at room temperature and an additional 15 min after adding antibodies. Incubations were done at room temperature according to the manufacturer's instructions (LightShift Chemiluminescent EMSA kit, cat no: 20148, Thermo Fisher Scientific).

**siRNA experiments**. siRNAs against *IRF4* were purchased from Qiagen (cat no. FlexiTube GeneSolution GS3662, IRF4), against *POU2F1* from Sigma (cat no. SASI_Hs01_00018404). L363 cells ($3 \times 10^6$) were transfected with 300 nmol siRNA using the Neon system (Thermo Fisher Scientific) in 100 µl volume. Electroporation conditions were 1500 V, 10 pulse width, and 3 pulse number. Luciferase analysis was done 24 h after transfection. In parallel, cells were harvested for immunoblotting. Luciferase constructs (5 µg plasmid/$3 \times 10^6$ cells) for *WAC* rs2790444 and *SMARCD3* rs78740585 reference and alternative alleles (Supplementary Table 5) were co-transfected with siRNA in a 100 µl reaction volume. The final siRNA concentration was 300 nmol/100 µl reaction mix. For immunoblotting, cells were lysed in 2X-Laemmli buffer (cat no: 161-0737; Bio-Rad) and sonicated for ten cycles of 30″/30″ s on/off on Bioruptor Pico (Diagenode). Quantitative measurement of total protein was done and 20 µg was loaded on 4 to 20% mini-PROTEAN TGX Gel (cat no: 456-1093, Bio-Rad). Post-electrophoresis gel was transferred to Trans-Blot turbo PVDF membrane and blotting was performed on Trans-Blot Turbo transfer system (Bio-Rad) using the same antibodies used in EMSA experiments.

**CRISPR/Cas9 deletion of variant-harboring regions**. Dual-sgRNA CRISPR/Cas9 deletion of variant-harboring regions is frequently used to investigate if a given genomic region (e.g., an intronic or distant enhancer) is involved in the transcriptional regulation of a given gene. Compared to CRISPR/Cas9 homology-directed repair (CRISPR-HDR), dual-sgRNA deletion has advantages in that it has high editing efficiency, and is applicable in a broader range of situations, as it does not require an effective sgRNA in the immediate vicinity of the variant (within a few base pairs). Here, we used dual-sgRNA to demonstrate functional couplings between variant-harboring regions in *WAC*, *ELL2*, and *CDCA7L* because the variants of interest themselves were not accessible to CRISPR-HDR due to a lack of efficient sgRNAs that cut DNA close to these variants.

To delete variant-harboring regions, we used a dual-sgRNA CRISPR/Cas9 in plasma cell lines. We identified functional sgRNAs targeting rs2790444, rs3777189, rs3777182-rs3777183, and rs4487645 regions; (Supplementary Table 8). The sgRNAs were cloned into the pSpCas9(BB)-2A-GFP PX458 vector (gift from Feng Zhang; Addgene cat no. 48138). Cloned sgRNA pairs were co-transfected using the Neon system (Thermo Fisher Scientific) into the following cell lines, which carry at least one copy of the high-expressing allele of the respective variants: RPMI-8226 (heterozygous for rs3777189 and rs3777183-rs3777182; DSMZ ACC402), OPM2 (homozygous for rs4487645-C; DSMZ ACC50), or MOLP8

(heterozygous for rs2790444; DSMZ ACC569). The cell lines were genotyped for the CRISPR-deleted variants and were found to have the expected genotype, as compared to data in the Cancer Cell Line Encyclopedia. The cell lines were not tested for mycoplasma. At 24 h post-transfection, GFP-positive cells were isolated using fluorescence-activated cell sorting. Genomic DNA was extracted and the targeted region was amplified by PCR to verify deletion (Supplementary Table 9). In parallel, RNA was prepared, reverse-transcribed, and quantified using SYBR Green qPCR assays (iTaq Univeral SYBR Green Supermix, cat no: 1725120; Supplementary Table 10).

**CRISPR/Cas9 with homology-directed repair**. To further test variant causality, we considered the possibility of précising-editing the identified MPRA-functional variants in MM cell lines using CRISPR/Cas9 with homology-directed repair (HDR)[74]. We achieved successful editing of *CDCA7L* rs4487645[C > A] in L363 cells. To generate L363 clones with different rs4487645 genotypes, we used the sgRNA sequence CCTCTGAAACTTACAATTCA with PAM sequence AGG cloned into a pSpCas9(BB)-2A-GFP vector (PX458, Addgene), along with the following repair templates: GTTGACCTATAAGGAAGCTGGCTCACAGAG GCTAGGGACAGATGAACCTCTTCGATAAAATTAAGAGA[G/T]AAGTG AAACCTTGAATTGTAAGTTTCAGAGGCTGCTTAAAGGGGACCAGGAG AATGGAGTAGAGAGCATAGCCTCAGTGTAA. Repair templates were synthesized by IDT (Alt-R HDR donor oligo, 2 nmol), with IDTs proprietary for 5′ and 3′ end modification for increased stability in the cell post-transfection. Plasmid and repair templates were co-transfected into L363 cells using a Neon electroporation system (Thermo Fisher). Post 48 h of transfection, single GFP positive cells were sorted using a BD FACSAria Fusion and cultured in a 96 well plate. Clones were genotyped for rs4487645 using Taqman genotyping assay (C_26972688_10, part no. 4351379) on a StepOnePlus qPCR instrument (Applied Biosystems). The selected clones were also analyzed by Sanger sequencing of the region encompassing CRISPR edit by amplifying with primers CDCA7L_F and CDCA7L_R (Supplementary Table 10). Because L363 is a genetically unstable cell line, and because CRISPR editing may introduce local DNA copy number changes due to chromothripsis, we also measured the *CDCA7L* DNA copy number in each clone using the Taqman copy number assay (Hs 02885634_cn; cat no. 4400291) with reference assay (RNasep, cat no. 4403326) in a duplex qPCR setup (Applied Biosystems, StepOnePlus). To calibrate the assay, we used DNA from two healthy blood donors. Copy numbers were calculated using CopyCaller v2.1 software (Thermo Fisher). To quantify *CDCA7L* expression, we used qPCR (Supplementary Table 10) with iTaq universal SYBR master mix (cat no. 1725120, Bio-Rad) and *GAPDH* as endogenous reference genes. To test for association between CRISPR-edited rs4487645 genotype and *CDCA7L* expression (quantified as $2^{-\Delta Ct}$ relative to *GADPH*), we used multivariate regression with *CDCA7L* DNA copy number as a covariate.

**caQTL data generation**. We generated ATAC-seq libraries from 50,000 $CD138^+$ magnetic bead-isolated MM plasma cells per sample using a protocol based on ref. [75]. Samples were obtained from the Norwegian MM Biobank in Trondheim, subject to ethical approval (Norway REK2014/97; Sweden 2019-06386). Libraries were prepared using the Nextera DNA Library Prep kit and sequenced (Illumina 2 × 125 bp). Adapter sequences in the ATAC-seq reads were removed using Trimmomatic (v0.36)[76] and aligned using Bowtie2 to hg38. Duplicate and mitochondrial reads were filtered out using SAMtools[77] and Picard (http://broadinstitute.github.io/picard). Transposase cut-sites were extracted from the BAM files using BEDtools. Read start sites were adjusted to represent the center of transposon binding event[78]. For quality control, we calculated the enrichment of ATAC-seq reads at transcription start sites (TSS) of protein-coding RefSeq genes as in ENCODE (www.encodeproject.org/data-standards/terms/#enrichment). In short, the distribution of read depths across 2-kb windows centered at TSSs were normalized by the average read depths in the flanking 100 bp on both ends. The average score across all genes was used as a TSS enrichment score, and we excluded samples that had an enrichment score <3.

**caQTL detection**. We estimated the local ATAC-seq signal intensity as the Tn5 cut-site density (i.e., the average number of cut-sites per bp) across a 150-bp sliding window positioned at every 10 bp across each LD region, normalized by the Tn5 cut-side density across the entire LD region in the same sample. Notably, the cut-side density quantity can be calculated across the LD region, as it is independent of specific nucleotides being present in the ATAC-seq sequences.

To identify caQTLs, we developed two computational approaches. First, we scanned the local ATAC-seq intensities for Pearson correlation with the MM lead variant for the LD block. Second, as a complementary approach inspired by methods previously developed by our lab[79–83], we developed asegmentation tool ("caQTLseg") to partition a region of LD into subregions with either lead variant-dependent or allele-independent local ATAC-seq signal intensity (link to a software in Code Availability section). In short, caQTLseg, which was inspired by signal reconstruction tools previously developed by us[79–81], takes as input the local ATAC-seq intensities $d_{ij}$ for window i = 1, ..., I and sample j = 1, ..., J. In an outer loop, we use dynamic programming to find a partitioning of the LD region that minimizes an inner cost function. At each step in the loop, the dynamic

programming algorithm suggests a candidate partitioning of the region consisting of a number of segments, whose breakpoints are shared across samples (as inherited variants can be assumed to have comparable effects across individuals). Given a candidate partitioning that consists of a number of segments s = 1, ..., S, caQTLseg then finds the values $f_{ij}(s)$ that minimize the sum of $L^2$ residuals $\lambda^{(s)} \times || d_{ij}^{(s)} - f_{ij}^{(s)}||^2$ across all i in the segment, averaged across all samples. Thus, caQTLseg seeks the partitioning and $f_{ij}(s)$ values that minimize

$$\sum_{s=1}^{S}\left(\lambda_2 + \frac{1}{J}\sum_{j=1}^{J}\sum_{i=i0(s)}^{i1(s)} \lambda^{(s)}\left\|d_{ij}^{(s)} - f_{ij}(s)\right\|^2\right) \qquad (1)$$

where $i0(s)$ and $i1(s)$ are the indices of the first and last windows of each segment. In every second segment, caQTLseg alternatingly fits allele-independent and allele-independent $f_{ij}(s)$ values. In allele-independent segments, the optimal $f_{ij}(s)$ values are the average of the $d_{ij}(s)$ across all j (i.e., the same optimal values are fit to all samples). In this case, caQTL also sets $\lambda^{(s)} = 1$. In allele-dependent segments, caQTLseg fits a linear model $f_{ij}(s) = a_i^{(s)} + g_j \times b_i^{(s)}$, where $a_i^{(s)}$ and $b_i^{(s)}$ are shared across all j, and $g_j$ is the variant genotype of sample j. In this case, $\lambda^{(s)}$ is set to a prespecified parameter $\lambda_1 > 1$ that serves to calibrate the cost of an allele-dependent model against the cost of an allele-independent model. Since an allele-dependent model is more flexible than an allele-independent model, it will always produce a lower $L^2$ residual, and $\lambda_1$ must therefore be greater than 1 in order to prevent the dynamic programming algorithm from always choosing the allele-dependent model. Increasing $\lambda_1$ makes it more difficult to call a segment allele-dependent, yielding a more conservative segmentation. Following the computation of the segment-specific cost, the total cost for the partitioning is calculated as the sum of segment-specific costs plus an additional regularization penalty calculated as the number of segments multiplied by a prespecified parameter $\lambda_2 > 0$ that determines the degree of over-segmentation versus under-segmentation. Increasing $\lambda_2$ produces a solution with fewer segments.

To estimate the noise level, we defined a statistic π0, defined as $min(n_0/n_1, 1)$, where $n_0$ is the average number of base pairs in the region-of-interest that are called allele-dependent under the null (i.e., when genotypes are randomly permuted between samples) and $n_1$ is the number of base pairs in the region-of-interest that are called allele-dependent with correctly assigned, unpermuted genotypes. We calculated $n_0$ using 500 random genotype permutations. The π0 statistic serves to estimate the proportion of signal that can be attributed to noise (similar to, say, the false discovery rate), and can be used to titrate $\lambda_1$ and $\lambda_2$. Clearly, π0 approaches 0 when $\lambda_1$ and $\lambda_2$ increase (more conservative segmentation) and 1 when $\lambda_1$ and $\lambda_2$ approach 1 and 0, respectively (less conservative segmentation).

To identify caQTLs, we used a two-stage approach, with a discovery set of 56 samples and a follow-up set of 105 samples. In the caQTLseg analysis, we used $\lambda_1 = 1.075$ and $\lambda_2 = 10^{-1.5}$. With these parameters, we detected allele-dependent regions conservatively in the combined data set of 161 samples (π0 = 0.03 for *SMARCD3* rs78740585; π0 = 0.0022 for *CDCA7L* rs4487645; π0 = 0.0022 for *CEP120* rs6595443). To further assess the robustness of the results, we also repeated the analysis with a broad range of parameter choices ($\lambda_1$ from 1.025 to 1.20, $\lambda_2$ from $10^{-1}$ to $10^{-5}$). Throughout, we identified essentially the same regions as allele-dependent, though the estimated noise level (π0) and the degree of fragmentation varied as expected (Supplementary Table 11 and Supplementary Figs. 12, 13).

**Statistics and reproducibility**. The experiments in Fig. 5c were done three times, Fig. 5e once, Fig. 6b twice, Fig. 6d once, and Fig. 6f twice. The gels shown are representative. Agreeing results were seen in the replicates.

**Reporting Summary**. Further information on research design is available in the Nature Research Reporting Summary linked to this article.

## Data availability

The raw sequencing data for the MPRA experiment have been deposited in the Sequence Read Archive, accession no. PRJNA679966 and are publicly available. The ATAC-sequencing data for primary $CD138^+$ MM plasma cells have been deposited in the European Genome-phenome Archive (EGA), accession no. EGAS00001005394 and EGAD00001007814 and are available to other researchers with controlled access. The PCHi-C data for KMS11 is available through EGA; accession number EGAS000010026 14 and EGAD00001003597 and are available to other researchers with controlled access, as are the meQTL data (accession number EGAS00001005788 and EGAD00010002259). The following previously published data sets were used: Gene expression data for MM samples from the CoMMMPASS study, available in dbGaP, accession number phs000748.v7.p4 (available to senior investigators through authorized access after application in dbGaP)[https://www.ncbi.nlm.nih.gov/projects/gap/cgi-bin/study.cgi?study_id=phs000748.v7.p4]; publicly available blood eQTL data from the eQTLGen Consortium[http://www.eqtlgen.org]; and publicly available gene expression data from the NCBI Gene Expression Omnibus (GEO) repository, accession numbers GSE111199, GSE24759, GSE15695, GSE4581, GSE19784, GSE26760, and GSE5900. Source data are provided with this paper.

**13**

## Code availability

The source code (C++) for caQTLseg is available at GitHub[https://github.com/abhisheknrl/caQTLseg][84].

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

## Acknowledgements

This work was supported by grants from the Knut and Alice Wallenberg Foundation (2012.0193 and 2017.0436), the Swedish Research Council (2017-02023 and 2018-00424), the Swedish Cancer Society (2017/265), the Nordic Cancer Union (R217-A13329-18-S65), Arne and Inga-Britt Lundberg's Stiftelse (2017-0055), European Research Council (EU-MSCA-COFUND grant no. 754299 and 847583) Myeloma UK and Cancer Research UK (C1298/A8362), The National Institute of Health (R01 DK103794 and R01HL146500), the New York Stem Cell Foundation, a gift from the Lodish Family to Boston Children's Hospital, and Mr. Ralph Stockwell. We thank Ellinor Johnsson for her assistance between 2011 and 2020. We are indebted to the patients who participated in the study.

## Author contributions

R.A., A.N., M.P., and B.N. designed the project. C.C., M.T., E.L.B., L.D.-L., A.L.d.L.P., I.J., T.R., U.T., V.S., and R.H. contributed to design. R.A., M.P., C.C., L.D.L., O.M., G.N., and K.G. performed experiments. S.K., A.S., M.K., K.A., N.M., N.W., K.H., H.G., A.F., I.J., T.R., F.v.R., A.W., U.T., V.G.S., K.S., and R.H. contributed data or samples. R.A., A.N., M.P., C.C., M.T., M.W., E.L.B., L.D.-L., A.L.d.L.P., T.O., N.U.-D., M.S., C.A.L., G.H.H., G.T., N.W., and B.N. carried out statistical analyses or analyzed the data. R.A., A.N., M.P., C.C., M.T., M.W., L.D.-L., A.L.d.L.P., R.H., and B.N. wrote the manuscript. All authors contributed to the final manuscript.

## Funding

## Competing interests

Authors T.O., O.M., G.H.H., G.T., G.L.N., K.G., I.J., T.R., U.T., and K.S. are employed by deCODE Genetics/Amgen Inc. The remaining authors declare no competing interests.
