## [Peer Review File · Nature Communications]

REVIEWER COMMENTS

Reviewer #1 (Remarks to the Author): Expert in MPRA and eQTLs

The authors have performed a comprehensive set of analysis led by the MPRA in plasma cells, followed by caQTL. In general, the paper is well written with clear biological implication of the results on transcription factor binding site (TFBS)-altering variants that are detected by the MPRA experiments as well as the in vivo caQTL that can recover some of the in vitro MPRA hits. I have a few comments.

1. Can authors demonstrate the predictive performance of their MPRA hits in terms of patient outcomes? For example, do the MPRA hits significant improves polygenic risk scores compared to the 1039 GWAS hits of MM?
2. How about survival analysis to demonstrates significant Kaplan-Meier p-values for the GWASMPRA hits compared to the GWAS hits alone?
3. Authors mention about fine-map probability of causality using a fairly early approach by Wakefield (ref 60). I recommend that authors try more recent fine-mapping methods including FINEMAP, PAINTOR, RiVIERA to properly account for LD and functional annotations.
4. For the proposed caQTLseg algorithm, there is no computational analysis that demonstrates the validity of the approach. The description of the dynamic programming algorithm coupled with the linear regression to get $f_{ij}(s)$ is also very ambiguous. From what I understand, the more segments the lower the residual error in the regression.
 - a. How to regularize this to get minimal number of segments that best explain the data with some L2 penalty?
 - b. Simulations and/or functional enrichments and/or replication study are needed to demonstrate that caQTLseg is a robust approach.

Reviewer #2 (Remarks to the Author): Expert in multiple myeloma and ATAC-seq

In this manuscript by the leaders in the field of heritable risk in the blood cancer multiple myeloma, the authors attempt further refinement of myeloma risk alleles which they catalogued and validated by chromatin status profiling and Hi-C or similar in several previous publications including two previous publications in Nat Communications (PMID: 30213928; PMID: 27882933).

They employ different complementary methodologies aiming to pinpoint the causal variant(s) amongst several in the same block of linkage disequilibrium (LD).

Of note, the candidate target genes of the risk alleles had been already identified in their previous work and at least in the case of CDCA7L, the link between risk variant and gene regulation was previously published by authors of the present manuscript (PMID: 27882933).

Through their MPRA screening that involves replicate assays in two myeloma cell lines the authors identify 8 high significance SNP that are predicted to have the most impact on the transcriptional activity of their target genes.

As mentioned, the risk variant for CDCA7L had already been comprehensively dissected in a previous Nat Comms manuscript from the same group (PMID: 27882933). The additional assay in support of its regulatory significance comprises dual-sgRNA CRISPR/Cas9 deletion of the intronic region surrounding the risk allele.

For the rest of the risk variants a mixture of chromatin long range interaction assays, EMSA, luciferase assays in conjunction with candidate transcription factor knock down, dual-sgRNA CRISPR/Cas9 deletion of the region surrounding the risk variant and ATAC-seq chromatin accessibility signal intensity (caQTL) in broad regions surrounding the risk variants are employed.

While all these are supportive of the roles of the risk variants, they do not provide definitive proof of their regulatory role. This would require introducing risk vs non-risk variants into myeloma cells with the appropriate genetic background and ideally should include variants from the same in cis LD region that are not predicted by MPRA to have a regulatory role upon the gene of interest. Impact

on gene transcription, long range interactions and binding of predicted TF can then be evaluated.
Additional comments:

1. In Figure 4, chromatin long range interactions are shown only in the direction of the variant of interest and within a restricted genomic window. Additional long range interactions between the target gene promoter and other distal regions in either direction are possible; therefore, wider genomic windows should be also shown.
2. In Figure 4b, the risk variant is so close to the promoter that the long range interaction shown might just occur because of the inability of the assay to reliably resolve due to the proximity of the two genomic loci.
3. In Fig 4d there are no long range interactions between the CDCA7L promoter, a finding contrary to other assays in the previous publication by the authors; what is the explanation for this discrepancy?
4. Suppl Fig 6 & 7: How can it be ascertained that decreased transcription of the target gene following dual-sgRNA CRISPR/Cas9 deletion of the region surrounding the risk allele is not due to removal of the risk variant but to the fact that deletion of intronic regions of 100bp might themselves impact gene transcription in manner independent of the risk variant they contain?
5. What is the ChrHMM status of the genomic regions containing the risk variants?

Minor points

1. All 8 risk variants but one appear to be in introns of the gene they are predicted to regulate. Can the authors comment on the potential significance of this finding?
2. A somewhat more nuanced discussion of the significance of the findings of the paper for the biology of myeloma is required.

Reviewer #3 (Remarks to the Author): Expert in non coding variants, ATAC-seq, and CRISPR-Cas9 assays

In this manuscript, Ajore et al used massively parallel reporter assays (MPRA) to screen 1,039 variants associated with multiple myeloma (MM). By the nature of these assay, the outcome is prioritization of potential variants that may have an impact on the gene expression. Although the assay is not performed on the endogenous gene loci, the approach is powerful for the initial screening purposes. The authors then complemented their initial findings with targeted luciferase and locus-specific manipulations to focus on a set of variants that alters the expression of a handful of genes.

The findings from this study and overall conclusions are based on strong premises. The manuscript is also well written. However, there are some noticeable weaknesses in the way the data is presented in this manuscript. Below, I will summaries some of these points.

On my initial reading the manuscript and the figures, I was extremely under impressed with the novelty of the data presented in the main figures. However, after second reading and going through the details of the supplementary figures, I realized that the most critical and novel data is, for some reason, is presented in the supplementary figures. Therefore, the manuscript is emphasizing some analysis in the main figures which are not novel and not worthy. The more important novel findings are all included in the supplementary figures. For example; Figure 1 and Figure 2 do not contain any novel data. They are just showing what is already known or expected. I understand that Figure 1 a and b may be needed to show the strategy but Fig1c and b can be added to supplementary figures as these are just crude analysis of the barcode counts.

On the other hand, the most critical data where this manuscript is contributing tp the field is the potential molecular findings on why certain variants are effective. For some reasons these critical data is presented in the supplementary figures. For instance, the data presented in Supplementary Figure 1, Sup. Figure 3, Sup. Figure 4 and Sup. Figure 5, 7 and 7 are the more critical and exciting data than almost all main figures. These figures need to be highlighted in the main figures.

Here are some minor weaknesses.

- o The texts in certain figures are too small to read. Please increase the font size.
- o In Figure 5, I assume the “cut site intensity” is meant for ATAC-Seq signal intensity? Please use a name that is more broadly understand.
- o In the same figure, the analysis is needed to compare the signal intensity of each of the alleles. With the current presentation it is hard to see whether the difference in ATAC-Seq intensity is significant or not.
- o Figure 4 is displaying potentially a Hi-C data, but it is not clear whether these “looping interactions” are significant or not. What are the red, yellow bars mean? Are they just showing a single paired-end Hi-C reads? How do we know that this is significant?
- o Some explanation or discussion is needed for the discrepancy between the number of significant variants detected in L363 and MOLP8 cells. Is it possible that MOLP8 is not from the same lineage?
- o P-values are needed for the data presented in supplementary figure 3.
- o Please show the full gel image for the EMSA results so that the equal loadings can be assessed.

Response to comments

We thank the referees for their constructive comments. All points have now been addressed. The changes are listed below, and also indicated in blue in the revised manuscript.

Response to comments from Reviewer #1

1.1. “The authors have performed a comprehensive set of analysis led by the MPRA in plasma cells, followed by caQTL. In general, the paper is well written with clear biological implication of the results on transcription factor binding site (TFBS)-altering variants that are detected by the MPRA experiments as well as the in vivo caQTL that can recover some of the in vitro MPRA hits.”

Response: We thank the referees for their encouragement. We are pleased that they found our article of interest.

1.2. “For the proposed caQTLseg algorithm, there is no computational analysis that demonstrates the validity of the approach. The description of the dynamic programming algorithm coupled with the linear regression to get $f_{ij}(s)$ is also very ambiguous. From what I understand, the more segments the lower the residual error in the regression.(a) How to regularize this to get minimal number of segments that best explain the data with some L2 penalty? (b) Simulations and/or functional enrichments and/or replication study are needed to demonstrate that caQTLseg is a robust approach.”

Response: As requested, to clarify matters we now include additional information in the

Results

(page 10), **Methods** (page 31-33), and include a new **Supplementary Table 13, Supplementary**

Fig. 11 and 12. Regularized L_2 regression is widely employed to reconstruct signals from noisy data, and the behavior of such algorithms is well documented. The development of caQTL was inspired by regularized segmentation algorithms previously developed by us (PMID 18208590, PMID 19228802). The main difference is that caQTLseg finds a consensus segmentation of several input signals in parallel, instead of applying a standard segmentation to each input signal independently. The rationale for this is that the goal is to reconstruct the effects of a common genetic variant, and this effect can be assumed to be the same across all samples.

As the reviewer points out, the main question is how to select the λ_1 and λ_2 parameters, which jointly determine the balance between sensitivity (oversegmentation) and specificity (undersegmentation). The λ_1 parameter serves to calibrate the cost of an allele-dependent model against the cost of an allele-independent model. Since an allele-dependent model is more flexible than an allele-independent model, it will always produce a lower L_2 residual, and λ_1 must therefore be greater than 1 in order to prevent the dynamic programming algorithm from always choosing the allele-independent model. Increasing λ_1 makes it more difficult to call a segment allele-dependent, yielding a more conservative segmentation. The parameter λ_2 specifies the cost for adding an extra segment, and increasing λ_2 produces a solution with fewer segments. To select λ_1 and λ_2 , we developed a statistic, p_0 , which indicates how much of the region-of-interest that would be called allele-dependent under the null hypothesis, relative to how much was called allele-dependent in the actual data. We let $p_0 = \min(n_0/n_1, 1)$, where n_0 is the average number of base pairs in the region-of-interest that are called allele-dependent under the null (*i.e.*, when genotypes are randomly permuted between samples) and n_1 is the number of base pairs in the region-of-interest that are called allele-dependent with correctly assigned, unpermuted genotypes. We calculated p_0 by generating 500 random genotypes. p_0 serves to estimate the proportion of signal that can be attributed to noise (similar to, say, a false discovery rate), and can thus be used to find reasonable λ_1 and λ_2 . For example, a p_0 value of 0.05 means that, if there was no signal in the data, the total size of the regions called allele-dependent would be ~5% the size than the region that was called allele-dependent in the actual data, or, conversely, that 95% of the signal can be expected to be true signal. Clearly, p_0 will approach 0 when λ_1 and λ_2 increase (as such values force conservative segmentation), and 1 when λ_1 and λ_2 approach 1 and 0, respectively (as such values allow less conservative segmentation).

To identify caQTLs, we used a two-stage approach, with a discovery set of 56 samples (**Supplementary Fig. 10**) and a follow-up set of 105 samples (**Fig. 8**). In the caQTLseg analysis,

we used $\lambda_1=1.075$ and $\lambda_2=10^{-1.5}$. With these parameters, we detected allele-dependent regions at conservative p_0 values in the combined data set of 161 samples ($p_0 = 0.03$ for *SMARCD3* rs78740585; $p_0 = 0.0022$ for *CDC47L* rs4487645; $p_0 = 0.0022$ for *CEP120* rs6595443). To further address the reviewers requested, and to assess the robustness of the results, we also repeated the analysis with a broad range of parameter choices (λ_1 from 1.025 to 1.20; λ_2 from 10^{-1} to 10^{-5}) in order to demonstrate that our results were not a fluke of a particular choice of λ_1 and λ_2 values. Throughout, we identified essentially the same regions as allele-dependent, though the estimated noise level (p_0) and the degree of fragmentation varied as expected (**Supplementary Table 13** and **Supplementary Fig. 11** and **12**). In all, the caQTLseg is inspired by a well-known

class of signal reconstruction algorithms, p_0 allows estimation of the signal-to-noise ratio, the choice of λ_1 and λ_2 is not critical, and the results hold in a two-stage analysis.

1.3. "Authors mention about fine-map probability of causality using a fairly early approach by Wakefield

(ref 60). I recommend that authors try more recent fine-mapping methods including FINEMAP, PAINTOR, RiVIERA to properly account for LD and functional annotations."

Response: This is now addressed in the Methods section (page 25). We intentionally chose not to

use fine-mapping methods that factor in functional information, but instead a pure genetics method such that our downstream functional analyses would not be biased. Particularly, gchromVAR

cell type enrichment analysis relies on a pure genetics fine-mapping output to produce unbiased estimates of cell type enrichments. Further, we account for LD in a separate step by performing stepwise conditional analysis. In that analysis, we did not find any secondary signals at any of the loci. Thus, one of the major advantages of newer fine-mapping methods, which is the ability to identify multiple causal signals per region, was already accounted for through conditional analysis. Finally, previous studies have shown high correlations between approximate Bayes' fine-mapping and other fine-mapping methods (c.f., PMID 31578528).

1.4 “Can authors demonstrate the predictive performance of their MPRA hits in terms of patient outcomes? For example, do the MPRA hits significantly improve polygenic risk scores compared to the

1,039 GWAS hits of MM?”... “How about survival analysis to demonstrates significant Kaplan-Meier pvalues

for the GWAS-MPRA hits compared to the GWAS hits alone?”

Response: In this study, we are investigating the functional basis of risk loci for MM, not genetic

variants influencing patient outcome/survival. Hence, any such analysis is inappropriate.

Additionally, all of the variants that were evaluated by MPRA map to known MM risk loci and, as such, are in linkage disequilibrium (i.e. correlated) with the reported lead variants. In view of this, the performance of a polygenic risk score based on the MPRA-functional variants at each locus as compared with the lead variant will not be appreciably different.

Response to comments from Reviewer #2

2.1 “In Fig. 4, chromatin long range interactions are shown only in the direction of the variant of interest

and within a restricted genomic window. Additional long range interactions between the target gene

promoter and other distal regions in either direction are possible; therefore, wider genomic windows

should be also shown.”

Response: As requested, these data are provided in a new **Supplementary Figure 8**.

2.2. “In Fig. 4b, the risk variant is so close to the promoter that the long range interaction shown might

just occur because of the inability of the assay to resolve due to the proximity of the two genomic loci.”

Response: This is now clarified in **Results** (page 7) and in the legend of **Fig. 4** (page 17).

rs2790444 variant maps close to the WAC transcription start, within the PCHi-C bait region.

2.3. “How can it be ascertained that decreased transcription of the target gene following dual-sgRNA

CRISPR/Cas9 deletion of the region surrounding the risk allele is not due to removal of the risk variant

but to the fact that deletion of intronic regions of 100-bp might themselves impact gene transcription in

manner independent of the risk variant they contain?”

Response: We have revised our text to address this point in **Results** (page 8-9) and **Methods**

(page 28-29). Dual-sgRNA CRISPR/Cas9 deletion of variant-harboring regions is used to investigate if a given genomic region (e.g., an intronic or distant enhancer) is linked to the transcriptional regulation of a given target gene. Compared to CRISPR/Cas9 homology-directed repair (CRISPR-HDR), dual-sgRNA deletion has advantages in that it has high editing

efficiency, and is applicable in a broader range of situations, not requiring an effective sgRNA in the immediate vicinity of the variant. We successfully used dual-sgRNA to test for functional couplings between variant-harboring regions in *WAC*, *ELL2* and *CDCA7L*. Moreover we were also able to precision-edit the *CDCA7L* variant using CRISPR-HDR (addressing point 2.4).

2.4. *“While all these are supportive of the roles of the risk variants, they do not provide definitive proof of their regulatory role. This would require introducing risk vs non-risk variants into myeloma cells with the appropriate genetic background and ideally should include variants from the same in cis LD region that are not predicted by MPRA to have a regulatory role upon the gene of interest.”*

Response: Additional CRISPR-HDR data are now included in **Results** (page 9), a new **Fig. 7d**,

Discussion (page 11), and **Methods** (page 28-29) to clarify matters. While we acknowledge that

the functional dissection of a GWAS signal should ideally involve systematic editing of each variant within the LD block (for example using CRISPR-HDR or base editors), it is widely recognized that such an approach is not realistic using current methods. Aside from the sheer scale of the work, only some variants will be accessible to precision editing (as both CRISPRHDR

and base editors require an effective sgRNA sequence in the immediate vicinity of the variant; additionally base editors can only produce certain base changes). Moreover, and importantly in MM, it is not possible to culture primary plasma cells or primary multiple myeloma cells *ex vivo*, and thus any editing experiments will need to be done in cell lines.

For these reasons, we initially followed up our MPRA screen with dual-sgRNA CRISPR/Cas9 experiments to link variant-harboring regions to eQTL target genes. Such edits were achieved for the variants-harboring regions at *WAC*, *ELL2*, and *CDCA7L*. Additionally, during the revision, we successfully edited the rs4487645[C>A] variant at *CDCA7L* in L363 cells. We generated 6 rs4487645-C-homozygous, 3 rs4487645-A/C-heterozygous, and 6 rs4487645-A-homozygous single-cell clones. We observed a significant association between rs4487645 genotype and *CDCA7L* expression. Consistent with the MPRA data, the eQTL and the other functional data, the C allele associated with higher *CDCA7L* expression. These new data further support that rs4487645 underlies the *CDCA7L* eQTL (new **Fig. 7d**). For the other variants-of-interest, CRISPR editing was not achieved.

Finally, as a complement to our *in vitro* experiments, we performed caQTL analysis in primary MM plasma cells from a substantial series of patients (n=161), revealing the presence of allele-dependent regulatory activity at the precise positions of the *SMARCD3*, *CDCA7L* and *CEP120* MPRA-functional variants in an endogenous chromosomal context *in vivo* (**Fig. 8**).

2.5 *“What is the ChrHMM status of the genomic regions containing the risk variants?”*

Response: We now provide ChromHMM annotations for the KMS11 cell line in **Figure 4**, with

associated changes in the figure legend (page 17) and **Methods** (page 26).

2.6. *“All 8 risk variants but one appear to be in introns of the gene they are predicted to regulate. Can*

the authors comment on the potential significance of this finding?”

Response: This is now commented in **Discussion** (page 10). The fact that most of the identified

putative causal variants map to introns of their target genes is consistent with findings from other functional studies GWAS signals.

2.7. “A somewhat more nuanced discussion of the significance of the findings of the paper for the biology of myeloma is required.”

Response: We have revised the **Discussion** section acknowledging this point (page 10-11).

2.8 “In Fig. 4d there are no long range interactions between the CDCA7L promoter, a finding contrary to other assays in the previous publication by the authors. What is the explanation for this discrepancy?”

Response: The interaction with the *CDCA7L* promoter in our previous work was based on 3C analysis, a technique optimally suited for identifying short-range-defined interactions. PChi-C is partly agnostic (only the promoter being defined), in contrast to 3C and its effectiveness is governed by a number factors including restriction enzyme and sequencing depth, which can limit its ability to universally demonstrate interactions.

Response to comments from Reviewer #3

3.1. “In this manuscript, Ajore et al used massively parallel reporter assays (MPRA) to screen 1,039

variants associated with multiple myeloma (MM). The outcome is prioritization of potential variants that

may have an impact on the gene expression. Although the assay is not performed on the endogenous gene

loci, the approach is powerful for the initial screening purposes. The findings from this study and overall

conclusions are based on strong premises. The manuscript is also well written.”

Response: As per 1.1 we thank the referees for their encouragement, and are pleased that they found our article of interest.

3.2. “The most critical and novel data is presented in the supplementary figures.” ... “I understand that

Figure 1 a and b may be needed to show the strategy but Fig1c and b can be added to supplementary

figures as these are just crude analysis of the barcode counts.” ... “The most critical data is the potential

molecular findings on why certain variants are effective. For some reasons these critical data is presented in the supplementary figures. For instance, the data presented in Supplementary Figure 1, Sup.

Figure 3, Sup. Figure 4 and Sup. Figure 5, 6 and 7 are the more critical and exciting data than almost all

main figures. These figures need to be highlighted in the main figures.”

Response: We acknowledge this point and have rearranged the figures accordingly. Firstly, the identification of target cell types (**Fig. 2**) has been moved to the supplements (**Supplementary Fig. 1 to 3**). Secondly, the individual barcode scores in L363 cells for the eight selected variants are now shown in a new **Fig. 3**, to highlight the most central MPRA data. Additionally, the complete set of individual barcode plots for all MPRA-functional variants in both L363 and MOLP8 cells is shown in **Supplementary Fig. 6**. Thirdly, the functional data for the *SMARCD3*

and *WAC* variants have been moved from the supplements to a new **Fig. 5** and new **Fig. 6**.

Fourthly, the CRISPR/Cas9 data for *ELL2* and *CDCA7L* have been moved from the supplements

to a new **Fig. 7**. We prefer to keep **Fig. 1c** and **1d** in the main figures as we feel these illustrate key aspects of the MPRA experiment. We also prefer to keep the detailed luciferase data plots in the supplements (**Supplementary Fig. 4**), as these data are summarized in **Fig. 2c**.

3.3. *"The texts in certain figures are too small to read. Please increase the font size."*

Response: As requested these changes have been made.

3.5. *"In Fig. 5, I assume the "cut site intensity" is meant for ATAC-seq signal intensity? Please use a name that is more broadly understood."*

Response: As requested, rather than use the phrase "cut site intensity", we now use the phrase "ATAC-seq signal intensity".

3.6. *"In the same figure, analysis is needed to compare the signal intensity of each of the alleles. With the current presentation it is hard to see whether the difference in ATAC-Seq intensity is significant or not."*

Response: Legend for **Fig. 8** (page 21) revised for clarification. The difference in ATAC-seq signal intensity between genotype groups is shown by the false discovery rate (Q-value) for the Pearson correlation between genotype and ATAC-seq signal at each position (dashed line).

3.7. *"Figure 4 is displaying Hi-C data, but it is not clear whether these interactions are significant."*

Response: Significance criteria clarified in **Methods** (page 26) and **Figure 4** legend (page 17).

3.8. *"Some explanation or discussion is needed for the discrepancy between the number of significant variants detected in L363 and MOLP8 cells. Is it possible that MOLP8 is not from the same lineage?"*

Response: This is now discussed in **Results** (page 6). Both L363 and MOLP8 are MM plasma cell lines. The higher number of significant variants in L363, compared to MOLP8, cells is congruent with a higher transfection efficiency (54% for L363 versus 15% for MOLP8) and higher post-transfection viability (90% for L363 versus 65% for MOLP8).

3.9. *"P-values are needed for the data presented in Supplementary Fig. 3."*

Response: P-values are now provided.

3.10. *"Please show the full gel image for the EMSA results so that the equal loadings can be assessed."*

Response: The full gel images are now provided in a new **Supplementary Fig. 7** and **9**.

REVIEWER COMMENTS

Reviewer #1 (Remarks to the Author): Expert in MPRA and eQTLs

The authors have performed a comprehensive set of analysis led by the MPRA in plasma cells, followed by caQTL. In general, the paper is well written with clear biological implication of the results on transcription factor binding site (TFBS)-altering variants that are detected by the MPRA experiments as well as the in vivo caQTL that can recover some of the in vitro MPRA hits. I have a few comments.

1. Can authors demonstrate the predictive performance of their MPRA hits in terms of patient outcomes? For example, do the MPRA hits significant improves polygenic risk scores compared to the 1039 GWAS hits of MM?
2. How about survival analysis to demonstrates significant Kaplan-Meier p-values for the GWASMPRA

hits compared to the GWAS hits alone?

3. Authors mention about fine-map probability of causality using a fairly early approach by Wakefield (ref 60). I recommend that authors try more recent fine-mapping methods including FINEMAP, PAINTOR, RiVIERA to properly account for LD and functional annotations.

4. For the proposed caQTLseg algorithm, there is no computational analysis that demonstrates the validity of the approach. The description of the dynamic programming algorithm coupled with the linear regression to get $f_{ij}(s)$ is also very ambiguous. From what I understand, the more segments the lower the residual error in the regression.

a. How to regularize this to get minimal number of segments that best explain the data with some L2 penalty?

b. Simulations and/or functional enrichments and/or replication study are needed to demonstrate that caQTLseg is a robust approach.

Reviewer #2 (Remarks to the Author): Expert in multiple myeloma and ATAC-seq

In this manuscript by the leaders in the field of heritable risk in the blood cancer multiple myeloma, the authors attempt further refinement of myeloma risk alleles which they catalogued and validated by chromatin status profiling and Hi-C or similar in several previous publications including two previous publications in Nat Communications (PMID: 30213928; PMID: 27882933).

They employ different complementary methodologies aiming to pinpoint the causal variant(s) amongst several in the same block of linkage disequilibrium (LD).

Of note, the candidate target genes of the risk alleles had been already identified in their previous work and at least in the case of CDCA7L, the link between risk variant and gene regulation was previously published by authors of the present manuscript (PMID: 27882933).

Through their MPRA screening that involves replicate assays in two myeloma cell lines the authors identify 8 high significance SNP that are predicted to have the most impact on the transcriptional activity of their target genes.

As mentioned, the risk variant for CDCA7L had already been comprehensively dissected in a previous Nat Comms manuscript from the same group (PMID: 27882933). The additional assay in support of its regulatory significance comprises dual-sgRNA CRISPR/Cas9 deletion of the intronic region surrounding the risk allele.

For the rest of the risk variants a mixture of chromatin long range interaction assays, EMSA, luciferase assays in conjunction with candidate transcription factor knock down, dual-sgRNA CRISPR/Cas9 deletion of the region surrounding the risk variant and ATAC-seq chromatin accessibility signal intensity (caQTL) in broad regions surrounding the risk variants are employed.

While all these are supportive of the roles of the risk variants, they do not provide definitive proof of their regulatory role. This would require introducing risk vs non-risk variants into myeloma cells with the appropriate genetic background and ideally should include variants from the same in cis LD region that are not predicted by MPRA to have a regulatory role upon the gene of interest. Impact on gene transcription, long range interactions and binding of predicted TF can then be evaluated.

Additional comments:

1. In Figure 4, chromatin long range interactions are shown only in the direction of the variant of interest and within a restricted genomic window. Additional long range interactions between the target gene promoter and other distal regions in either direction are possible; therefore, wider genomic windows should be also shown.

2. In Figure 4b, the risk variant is so close to the promoter that the long range interaction shown might just occur because of the inability of the assay to reliably resolve due to the proximity of the two genomic loci.

3. In Fig 4d there are no long range interactions between the CDCA7L promoter, a finding contrary to other assays in the previous publication by the authors; what is the explanation for this discrepancy?

4. Suppl Fig 6 & 7: How can it be ascertained that decreased transcription of the target gene following dual-sgRNA CRISPR/Cas9 deletion of the region surrounding the risk allele is not due to removal of the risk variant but to the fact that deletion of intronic regions of 100bp might themselves impact gene transcription in manner independent of the risk variant they contain?
5. What is the ChrHMM status of the genomic regions containing the risk variants?

Minor points

1. All 8 risk variants but one appear to be in introns of the gene they are predicted to regulate. Can the authors comment on the potential significance of this finding?
2. A somewhat more nuanced discussion of the significance of the findings of the paper for the biology of myeloma is required.

Reviewer #3 (Remarks to the Author): Expert in non coding variants, ATAC-seq, and CRISPR-Cas9 assays

In this manuscript, Ajore et al used massively parallel reporter assays (MPRA) to screen 1,039 variants associated with multiple myeloma (MM). By the nature of these assay, the outcome is prioritization of potential variants that may have an impact on the gene expression. Although the assay is not performed on the endogenous gene loci, the approach is powerful for the initial screening purposes. The authors then complemented their initial findings with targeted luciferase and locus-specific manipulations to focus on a set of variants that alters the expression of a handful of genes.

The findings from this study and overall conclusions are based on strong premises. The manuscript is also well written. However, there are some noticeable weaknesses in the way the data is presented in this manuscript. Below, I will summarize some of these points.

On my initial reading the manuscript and the figures, I was extremely under impressed with the novelty of the data presented in the main figures. However, after second reading and going through the details of the supplementary figures, I realized that the most critical and novel data is, for some reason, is presented in the supplementary figures. Therefore, the manuscript is emphasizing some analysis in the main figures which are not novel and not worthy. The more important novel findings are all included in the supplementary figures. For example; Figure 1 and Figure 2 do not contain any novel data. They are just showing what is already known or expected. I understand that Figure 1 a and b may be needed to show the strategy but Fig1c and b can be added to supplementary figures as these are just crude analysis of the barcode counts.

On the other hand, the most critical data where this manuscript is contributing to the field is the potential molecular findings on why certain variants are effective. For some reasons these critical data is presented in the supplementary figures. For instance, the data presented in Supplementary Figure 1, Sup. Figure 3, Sup. Figure 4 and Sup. Figure 5, 7 and 7 are the more critical and exciting data than almost all main figures. These figures need to be highlighted in the main figures.

Here are some minor weaknesses.

- o The texts in certain figures are too small to read. Please increase the font size.
- o In Figure 5, I assume the "cut site intensity" is meant for ATAC-Seq signal intensity? Please use a name that is more broadly understood.
- o In the same figure, the analysis is needed to compare the signal intensity of each of the alleles. With the current presentation it is hard to see whether the difference in ATAC-Seq intensity is significant or not.
- o Figure 4 is displaying potentially a Hi-C data, but it is not clear whether these "looping interactions" are significant or not. What are the red, yellow bars mean? Are they just showing a single paired-end Hi-C reads? How do we know that this is significant?
- o Some explanation or discussion is needed for the discrepancy between the number of significant variants detected in L363 and MOLP8 cells. Is it possible that MOLP8 is not from the same lineage?

- o P-values are needed for the data presented in supplementary figure 3.
- o Please show the full gel image for the EMSA results so that the equal loadings can be assessed.

Response to comments

We thank the referees for their constructive comments. All points have now been addressed. The changes are listed below, and also indicated in blue in the revised manuscript.

Response to comments from Reviewer #1

1.1. "The authors have performed a comprehensive set of analysis led by the MPRA in plasma cells, followed by caQTL. In general, the paper is well written with clear biological implication of the results on transcription factor binding site (TFBS)-altering variants that are detected by the MPRA experiments as well as the in vivo caQTL that can recover some of the in vitro MPRA hits."

Response: We thank the referees for their encouragement. We are pleased that they found our article of interest.

1.2. "For the proposed caQTLseg algorithm, there is no computational analysis that demonstrates the validity of the approach. The description of the dynamic programming algorithm coupled with the linear regression to get $f_{ij}(s)$ is also very ambiguous. From what I understand, the more segments the lower the residual error in the regression. (a) How to regularize this to get minimal number of segments that best explain the data with some L2 penalty? (b) Simulations and/or functional enrichments and/or replication study are needed to demonstrate that caQTLseg is a robust approach."

Response: As requested, to clarify matters we now include additional information in the

Results

(page 10), **Methods** (page 31-33), and include a new **Supplementary Table 13,**

Supplementary

Fig. 11 and **12.** Regularized L_2 regression is widely employed to reconstruct signals from noisy data, and the behavior of such algorithms is well documented. The development of caQTL was inspired by regularized segmentation algorithms previously developed by us (PMID 18208590, PMID 19228802). The main difference is that caQTLseg finds a consensus segmentation of several input signals in parallel, instead of applying a standard segmentation to each input signal independently. The rationale for this is that the goal is to reconstruct the effects of a common genetic variant, and this effect can be assumed to be the same across all samples.

As the reviewer points out, the main question is how to select the λ_1 and λ_2 parameters, which jointly determine the balance between sensitivity (oversegmentation) and specificity (undersegmentation). The λ_1 parameter serves to calibrate the cost of an allele-dependent model against the cost of an allele-independent model. Since an allele-dependent model is more flexible than an allele-independent model, it will always produce a lower L_2 residual, and λ_1 must therefore be greater than 1 in order to prevent the dynamic programming algorithm from always choosing the allele-independent model. Increasing λ_1 makes it more difficult to call a segment allele-dependent, yielding a more conservative segmentation. The parameter λ_2 specifies the cost for adding an extra segment, and increasing λ_2 produces a solution with fewer segments.

To select λ_1 and λ_2 , we developed a statistic, p_0 , which indicates how much of the regionof-

interest that would be called allele-dependent under the null hypothesis, relative to how much was called allele-dependent in the actual data. We let $p_0 = \min(n_0/n_1, 1)$, where n_0 is the average number of base pairs in the region-of-interest that are called allele-dependent under the null (*i.e.*, when genotypes are randomly permuted between samples) and n_1 is the number of base pairs in the region-of-interest that are called allele-dependent with correctly assigned, unpermuted genotypes. We calculated p_0 by generating 500 random genotypes. p_0 serves to estimate the proportion of signal that can be attributed to noise (similar to, say, a false discovery rate), and can thus be used to find reasonable λ_1 and λ_2 . For example, a p_0 value of 0.05 means that, if there was no signal in the data, the total size of the regions called allele-dependent would be ~5% the size than the region that was called allele-dependent in the actual data, or, conversely, that 95% of the signal can be expected to be true signal. Clearly, p_0 will approach 0 when λ_1 and λ_2 increase (as such values force conservative segmentation), and 1 when λ_1 and λ_2 approach 1 and 0, respectively (as such values allow less conservative segmentation).

To identify caQTLs, we used a two-stage approach, with a discovery set of 56 samples (**Supplementary Fig. 10**) and a follow-up set of 105 samples (**Fig. 8**). In the caQTLseg analysis,

we used $\lambda_1=1.075$ and $\lambda_2=10^{-1.5}$. With these parameters, we detected allele-dependent regions at conservative p_0 values in the combined data set of 161 samples ($p_0 = 0.03$ for *SMARCD3* rs78740585; $p_0 = 0.0022$ for *CDC47L* rs4487645; $p_0 = 0.0022$ for *CEP120* rs6595443). To further address the reviewers requested, and to assess the robustness of the results, we also repeated the analysis with a broad range of parameter choices (λ_1 from 1.025 to 1.20; λ_2 from 10^{-1} to 10^{-5}) in order to demonstrate that our results were not a fluke of a particular choice of λ_1 and λ_2 values. Throughout, we identified essentially the same regions as allele-dependent, though the estimated noise level (p_0) and the degree of fragmentation varied as expected (**Supplementary Table 13** and **Supplementary Fig. 11** and **12**). In all, the caQTLseg is inspired by a well-known

class of signal reconstruction algorithms, p_0 allows estimation of the signal-to-noise ratio, the choice of λ_1 and λ_2 is not critical, and the results hold in a two-stage analysis.

1.3. "Authors mention about fine-map probability of causality using a fairly early approach by Wakefield

(ref 60). I recommend that authors try more recent fine-mapping methods including FINEMAP, PAINTOR, RiVIERA to properly account for LD and functional annotations."

Response: This is now addressed in the Methods section (page 25). We intentionally chose not to

use fine-mapping methods that factor in functional information, but instead a pure genetics method such that our downstream functional analyses would not be biased. Particularly, gchromVAR

cell type enrichment analysis relies on a pure genetics fine-mapping output to produce unbiased estimates of cell type enrichments. Further, we account for LD in a separate step by performing stepwise conditional analysis. In that analysis, we did not find any secondary signals at any of the loci. Thus, one of the major advantages of newer fine-mapping methods, which is the ability to identify multiple causal signals per region, was already accounted for through conditional analysis. Finally, previous studies have shown high correlations between approximate Bayes' fine-mapping and other fine-mapping methods (*c.f.*, PMID 31578528).

1.4 "Can authors demonstrate the predictive performance of their MPRA hits in terms of patient outcomes? For example, do the MPRA hits significantly improve polygenic risk scores compared to the

1,039 GWAS hits of MM?”... “How about survival analysis to demonstrates significant Kaplan-Meier pvalues

for the GWAS-MPRA hits compared to the GWAS hits alone?”

Response: In this study, we are investigating the functional basis of risk loci for MM, not genetic

variants influencing patient outcome/survival. Hence, any such analysis is inappropriate.

Additionally, all of the variants that were evaluated by MPRA map to known MM risk loci and, as such, are in linkage disequilibrium (*i.e.* correlated) with the reported lead variants. In view of this, the performance of a polygenic risk score based on the MPRA-functional variants at each locus as compared with the lead variant will not be appreciably different.

Response to comments from Reviewer #2

2.1 “In Fig. 4, chromatin long range interactions are shown only in the direction of the variant of interest

and within a restricted genomic window. Additional long range interactions between the target gene

promoter and other distal regions in either direction are possible; therefore, wider genomic windows

should be also shown.”

Response: As requested, these data are provided in a new **Supplementary Figure 8**.

2.2. “In Fig. 4b, the risk variant is so close to the promoter that the long range interaction shown might

just occur because of the inability of the assay to resolve due to the proximity of the two genomic loci.”

Response: This is now clarified in **Results** (page 7) and in the legend of **Fig. 4** (page 17). rs2790444 variant maps close to the *WAC* transcription start, within the PCHi-C bait region.

2.3. “How can it be ascertained that decreased transcription of the target gene following dual-sgRNA

CRISPR/Cas9 deletion of the region surrounding the risk allele is not due to removal of the risk variant

but to the fact that deletion of intronic regions of 100-bp might themselves impact gene transcription in

manner independent of the risk variant they contain?”

Response: We have revised our text to address this point in **Results** (page 8-9) and **Methods**

(page 28-29). Dual-sgRNA CRISPR/Cas9 deletion of variant-harboring regions is used to investigate if a given genomic region (*e.g.*, an intronic or distant enhancer) is linked to the transcriptional regulation of a given target gene. Compared to CRISPR/Cas9 homology-directed repair (CRISPR-HDR), dual-sgRNA deletion has advantages in that it has high editing efficiency, and is applicable in a broader range of situations, not requiring an effective sgRNA in the immediate vicinity of the variant. We successfully used dual-sgRNA to test for functional couplings between variant-harboring regions in *WAC*, *ELL2* and *CDCA7L*. Moreover we were also able to precision-edit the *CDCA7L* variant using CRISPR-HDR (addressing point 2.4).

2.4. “While all these are supportive of the roles of the risk variants, they do not provide definitive proof of

their regulatory role. This would require introducing risk vs non-risk variants into myeloma cells with the

appropriate genetic background and ideally should include variants from the same in cis LD region that

are not predicted by MPRA to have a regulatory role upon the gene of interest.”

Response: Additional CRISPR-HDR data are now included in **Results** (page 9), a new **Fig. 7d**,

Discussion (page 11), and **Methods** (page 28-29) to clarify matters. While we acknowledge that

the functional dissection of a GWAS signal should ideally involve systematic editing of each variant within the LD block (for example using CRISPR-HDR or base editors), it is widely recognized that such an approach is not realistic using current methods. Aside from the sheer scale of the work, only some variants will be accessible to precision editing (as both CRISPRHDR

and base editors require an effective sgRNA sequence in the immediate vicinity of the variant; additionally base editors can only produce certain base changes). Moreover, and importantly in MM, it is not possible to culture primary plasma cells or primary multiple myeloma cells *ex vivo*, and thus any editing experiments will need to be done in cell lines.

For these reasons, we initially followed up our MPRA screen with dual-sgRNA

CRISPR/Cas9 experiments to link variant-harboring regions to eQTL target genes. Such edits were achieved for the variants-harboring regions at *WAC*, *ELL2*, and *CDCA7L*. Additionally, during the revision, we successfully edited the rs4487645[C>A] variant at *CDCA7L* in L363 cells. We generated 6 rs4487645-C-homozygous, 3 rs4487645-A/C-heterozygous, and 6 rs4487645-A-homozygous single-cell clones. We observed a significant association between rs4487645 genotype and *CDCA7L* expression. Consistent with the MPRA data, the eQTL and the other functional data, the C allele associated with higher *CDCA7L* expression. These new data further support that rs4487645 underlies the *CDCA7L* eQTL (new **Fig. 7d**). For the other variants-of-interest, CRISPR editing was not achieved.

Finally, as a complement to our *in vitro* experiments, we performed caQTL analysis in primary MM plasma cells from a substantial series of patients (n=161), revealing the presence of allele-dependent regulatory activity at the precise positions of the *SMARCD3*, *CDCA7L* and *CEP120* MPRA-functional variants in an endogenous chromosomal context *in vivo* (**Fig. 8**).

2.5 “What is the ChrHMM status of the genomic regions containing the risk variants?”

Response: We now provide ChromHMM annotations for the KMS11 cell line in **Figure 4**, with

associated changes in the figure legend (page 17) and **Methods** (page 26).

2.6. “All 8 risk variants but one appear to be in introns of the gene they are predicted to regulate. Can

the authors comment on the potential significance of this finding?”

Response: This is now commented in **Discussion** (page 10). The fact that most of the identified

putative causal variants map to introns of their target genes is consistent with findings from other functional studies GWAS signals.

2.7. “A somewhat more nuanced discussion of the significance of the findings of the paper for the biology of myeloma is required.”

Response: We have revised the **Discussion** section acknowledging this point (page 10-11).

2.8 “In Fig. 4d there are no long range interactions between the CDCA7L promoter, a finding contrary

to other assays in the previous publication by the authors. What is the explanation for this discrepancy?”

Response: The interaction with the *CDCA7L* promoter in our previous work was based on 3C

analysis, a technique optimally suited for identifying short-range-defined interactions. PChi-C is partly agnostic (only the promoter being defined), in contrast to 3C and its effectiveness is governed by a number of factors including restriction enzyme and sequencing depth, which can limit its ability to universally demonstrate interactions.

Response to comments from Reviewer #3

3.1. "In this manuscript, Ajore et al used massively parallel reporter assays (MPRA) to screen 1,039

variants associated with multiple myeloma (MM). The outcome is prioritization of potential variants that

may have an impact on the gene expression. Although the assay is not performed on the endogenous gene

loci, the approach is powerful for the initial screening purposes. The findings from this study and overall

conclusions are based on strong premises. The manuscript is also well written."

Response: As per 1.1 we thank the referees for their encouragement, and are pleased that they found our article of interest.

3.2. "The most critical and novel data is presented in the supplementary figures." ... "I understand that

Figure 1 a and b may be needed to show the strategy but Fig1c and b can be added to supplementary

figures as these are just crude analysis of the barcode counts." ... "The most critical data is the potential

molecular findings on why certain variants are effective. For some reasons these critical data is presented in the supplementary figures. For instance, the data presented in Supplementary Figure 1, Sup.

Figure 3, Sup. Figure 4 and Sup. Figure 5, 6 and 7 are the more critical and exciting data than almost all

main figures. These figures need to be highlighted in the main figures."

Response: We acknowledge this point and have rearranged the figures accordingly. Firstly, the identification of target cell types (**Fig. 2**) has been moved to the supplements (**Supplementary Fig. 1 to 3**). Secondly, the individual barcode scores in L363 cells for the eight selected variants are now shown in a new **Fig. 3**, to highlight the most central MPRA data. Additionally, the complete set of individual barcode plots for all MPRA-functional variants in both L363 and MOLP8 cells is shown in **Supplementary Fig. 6**. Thirdly, the functional data for the **SMARCD3**

and **WAC** variants have been moved from the supplements to a new **Fig. 5** and new **Fig. 6**.

Fourthly, the CRISPR/Cas9 data for **ELL2** and **CDCA7L** have been moved from the supplements

to a new **Fig. 7**. We prefer to keep **Fig. 1c** and **1d** in the main figures as we feel these illustrate key aspects of the MPRA experiment. We also prefer to keep the detailed luciferase data plots in the supplements (**Supplementary Fig. 4**), as these data are summarized in **Fig. 2c**.

3.3. "The texts in certain figures are too small to read. Please increase the font size."

Response: As requested these changes have been made.

3.5. "In Fig. 5, I assume the "cut site intensity" is meant for ATAC-seq signal intensity? Please use a

name that is more broadly understood."

Response: As requested, rather than use the phrase "cut site intensity", we now use the phrase "ATAC-seq signal intensity".

3.6. *“In the same figure, analysis is needed to compare the signal intensity of each of the alleles. With the current presentation it is hard to see whether the difference in ATAC-Seq intensity is significant or not.”*

Response: Legend for **Fig. 8** (page 21) revised for clarification. The difference in ATAC-seq signal intensity between genotype groups is shown by the false discovery rate (Q-value) for the Pearson correlation between genotype and ATAC-seq signal at each position (dashed line).

3.7. *“Figure 4 is displaying Hi-C data, but it is not clear whether these interactions are significant.”*

Response: Significance criteria clarified in **Methods** (page 26) and **Figure 4** legend (page 17).

3.8. *“Some explanation or discussion is needed for the discrepancy between the number of significant variants detected in L363 and MOLP8 cells. Is it possible that MOLP8 is not from the same lineage?”*

Response: This is now discussed in **Results** (page 6). Both L363 and MOLP8 are MM plasma cell lines. The higher number of significant variants in L363, compared to MOLP8, cells is congruent with a higher transfection efficiency (54% for L363 versus 15% for MOLP8) and higher post-transfection viability (90% for L363 versus 65% for MOLP8).

3.9. *“P-values are needed for the data presented in Supplementary Fig. 3.”*

Response: P-values are now provided.

3.10. *“Please show the full gel image for the EMSA results so that the equal loadings can be assessed.”*

Response: The full gel images are now provided in a new **Supplementary Fig. 7** and **9**.